# Almost Free: Self-concordance in Natural Exponential Families and an Application to Bandits

**Shuai Liu**
University of Alberta
shuailiu725@gmail.com

**Alex Ayoub**
University of Alberta
aayoub@ualberta.ca

**Flore Sentenac**
HEC Paris
sentenac@hec.fr

**Xiaoqi Tan**
University of Alberta
xiaoqi.tan@ualberta.ca

**Csaba Szepesvári**
University of Alberta
csaba.szepesvari@gmail.com

## Abstract

We prove that single-parameter natural exponential families with subexponential tails are self-concordant with polynomial-sized parameters. For subgaussian natural exponential families we establish an exact characterization of the growth rate of the self-concordance parameter. Applying these findings to bandits allows us to fill gaps in the literature: We show that optimistic algorithms for generalized linear bandits enjoy regret bounds that are both second-order (scale with the variance of the optimal arm's reward distribution) and free of an exponential dependence on the bound of the problem parameter in the leading term. To the best of our knowledge, ours is the first regret bound for generalized linear bandits with subexponential tails, broadening the class of problems to include Poisson, exponential and gamma bandits.

## 1 Introduction

Single-parameter natural exponential families (NEFs) [Mor82] abound in statistical applications [MN89; Bro86; WJ08; ML09]. In this paper we study properties of NEFs and in doing so we make two main contributions:

1. We study how tail properties of the base distribution of a NEF impose limits on the NEF: if the base distribution is subexponential (subgaussian), we show that the NEF is *self-concordant* with a stretch factor that grows inverse quadratically (respectively, linearly)

2. In generalized linear bandits whose reward distributions follow a NEF with subexponential base distribution, we show how this new result can be utilized to derive a novel second order regret bound whose leading term is free of exponential dependencies on the problem parameter — the first such result for this setting.

The class of distributions our results extend to includes: normal, Poisson, exponential, gamma and negative binomial distributions. Our findings *partially* address conjectures on whether generalized linear models with unbounded targets are self-concordant [Bac10; Fau+20]. This significantly generalizes the case when the targets are assumed to be bounded[1] [Mar19; OB21; Rus+21] and thus extends the applicability of the results therein.

---

[1]This assumption does not hold for distributions with unbounded support such as: normal, Poisson, exponential, gamma and negative binomial distributions.

38th Conference on Neural Information Processing Systems (NeurIPS 2024).

Self-concordance in NEFs turns out to be useful for both optimization and statistical estimation. The self-concordance property controls the remainder term, or approximation error, of a NEF's second-order Taylor expansion. This is useful in designing and analyzing both estimation and optimization methods. Historically the self-concordance property was first found to be useful for optimization [NN89; NT08; ST19] and later for statistical estimation [Bac10; Mar+19; OB21; BFR23]. In this paper, we will employ the self-concordance property in bandit problems [LS20] where it helps with controlling the error terms related to estimation.

Generalized linear bandits (GLBs) [Fil+10] has emerged as a standard framework for studying the role that nonlinear function approximation plays in decision making problems. Earlier works on GLBs (or its special case of of logistic bandits) [Fil+10; LLZ17; SPL23] naively approximates the nonlinear function with a linear first order Taylor expansion. This approach ends up paying a price in the leading term of the regret bound that is exponential in the size of the true underlying parameter. In logistic bandits, [Fau+20] were the first to exploit the self-concordance property of the Bernoulli distribution in order to get regret bounds free of an exponential dependence on the size of the problem parameter in the leading term. [Fau+20] use self-concordance to get a tighter second order Taylor expansion that better captures the curvature of the sigmoid function. Employing improved self-concordant analysis, [AFC21] get second-order regret bounds for logistic bandits and [Jan+24] extend these results to GLBs, under the assumption that the underlying reward distributions are self-concordant. We build upon this line of research and fill a gap in the bandit literature by designing and analyzing algorithms for GLBs with subexponential reward distributions. To the best of the authors' knowledge, our work considers the most general setting in the sense that all the previous works on GLBs considered subgaussian or bounded reward distributions while ours consider subexponential ones. For subgaussian rewards, these works [Fil+10; Jun+21; SPL23; Rus+21] still depend on $\kappa$. Note that [Rus+21], also employs self-concordance in GLBs but they assume bounded reward and focuses on addressing non-stationarity of the environment. In addition, their bounds also scale with $\kappa$ in the leading term, which can be exponentially large for logistic bandits. [Jan+24] consider a similar setting to ours. They assume the moment generating function of the base distribution $Q$ is defined over the entire real line. This implies that $Q$ does not have a tail as heavy as an exponential distribution hence less general than our setting. A concurrent work [LYJ24] develops novel confidence sets for self-concordant GLBs to improve the theoretical bounds while we prove that all GLBs with light-tailed base distribution are self-concordant. An interesting future direction would be applying their techniques to improve the bandit result of our work. [Saw+24] considers GLBs with bounded reward, which is known to be self-concordant, in a regime that only a limited number of decision policy updates is available.

The paper is organized as follows: In Section 2, we introduce single-parameter NEFs and review key properties relevant to our analysis. Section 3 demonstrates the self-concordance property of subexponential (subgaussian) NEFs, with a quadratic (respectively, linear) growth rate of the stretch factor. Additionally, we establish the tightness of the linear growth rate for subgaussian NEFs. In Section 4 we apply these findings to subexponential GLBs and derive novel second-order regret bounds devoid of exponential dependencies on the problem parameter in the leading term. Proofs omitted from the main text are provided in the appendix, except for those of well-known results, which are referenced accordingly.

## 2 Preliminaries

In this section we first introduce the notation we will use. We then introduce natural exponential families, review some of their basic properties and illustrate the concepts introduced by means of an example.

### 2.1 Notation

For a real-valued differentiable function $f$ defined over an open interval, we use $\dot{f}$, $\ddot{f}$ and $\dddot{f}$ to denote the first, second and third derivative of $f$. The set of reals is denoted by $\mathbb{R}$, the set of nonnegative reals by $\mathbb{R}_+$. For a set $\mathcal{U} \subseteq \mathbb{R}$, we denote its interior by $\mathcal{U}^\circ$. For real numbers $a, b$, we use $a \wedge b$ and $a \vee b$ to denote the $\min\{a, b\}$ and $\max\{a, b\}$, respectively. With $\phi$ a logical expression, $\mathbb{I}\{\phi\} = 1$ if $\phi$ evaluates to true and $\mathbb{I}\{\phi\} = 0$, otherwise. We use $f \lesssim g$ to indicate that $g$ dominates $f$ up to a constant factor over their common domain. For $S \subseteq \mathbb{R}$, $x \in \mathbb{R}$, we let $S \pm x$ denote the set

$\{s \pm x : s \in S\}$. A distribution over the reals is centered if it has zero mean. We use $\mathbb{P}()$ to denote the probability measure over the probability space that holds our random variables and we let $\mathbb{E}$ to denote the expectation corresponding to this measure. By $Y \sim Q$ we mean that the distribution of $Y$, a random variable, is $Q$.

## 2.2 Single-parameter NEFs

In this section we give our definitions for the NEF. We only consider single-parameter natural exponential families when the base distribution is defined over the reals. We follow the approach of the beautifully written monograph of [Bro86] that the reader is also referred to for any statements made about NEFs with no proofs.

Given a probability distribution $Q$ over $\mathbb{R}$ let $M_Q : \mathbb{R} \to \mathbb{R}_+ \cup \{+\infty\}$ denote its *moment generating function* (MGF):

$$M_Q(u) = \int \exp(uy) Q(dy), \qquad u \in \mathbb{R}.$$

We will find it convenient to also use the logarithm of the moment generating function, which is known as its *cumulant generating function* (CGF). We denote this by $\psi_Q : \mathbb{R} \to \mathbb{R} \cup \{+\infty\}$. Thus, $\psi_Q(u) = \log M_Q(u)$.

Let $\mathcal{U}_Q = \{u \in \mathbb{R} : M_Q(u) < \infty\}$ denote the domain of $M_Q$ (and, thus the domain of $\psi_Q$). As it is well-known, $\psi_Q$ is convex and hence $\mathcal{U}_Q$ is always an interval (which, trivially, always contains 0). For a subset of $\mathcal{U}_Q$, denoted by $\mathcal{U} \subseteq \mathcal{U}_Q$, we call $\mathcal{Q} = (Q_u)_{u \in \mathcal{U}}$ a natural exponential family (NEF) generated by $Q$ where for any $u \in \mathcal{U}$ we have

$$Q_u(dy) = \frac{1}{M_Q(u)} \exp(uy) Q(dy).$$

It follows that $Q_u$ is also a probability distribution over the reals for any $u \in \mathcal{U}$ by definition. An equivalent, useful form for $Q_u$ is $Q_u(dy) = \exp(uy - \psi_Q(u)) Q(dy)$. In applications, $u$ denotes an unknown parameter that is to be estimated based on observations from $Q_u$. Thus, $\mathcal{U}$ allows one to express extra restrictions on the admissible parameters beyond the limits imposed by $Q$. We call $\mathcal{U}$ the *parameter space*, and $\mathcal{U}_Q$ the *natural parameter space*.

The distributions $Q$, $Q_u$ and parameter $u$ are referred to as the *base distribution*, the (exponentially) *tilted (base) distribution* and the *tilting parameter*. An NEF is said to be *regular* when $\mathcal{U}_Q$ is open. It is easy to see that for any $u, u_0 \in \mathcal{U}_Q$, $Q_u = (Q_{u_0})_{u-u_0}$, where the distribution on the right-hand side stands for the tilt of $Q_{u_0}$ with parameter $u - u_0$. As such, up to a constant shift of the parameter space, in a regular NEF, one can always assume that $0 \in \mathcal{U}_Q^\circ$. In fact, the same can be assumed for the parameter set, as long as $\mathcal{U}^\circ$ is nonempty. If $\mathcal{U}$ is an interval then $\mathcal{U}^\circ = \emptyset$ means that $\mathcal{U}$ is a singleton: An uninteresting case if we want to model a host of *non-identical* distributions.

In a regular family, an equivalent way to parameterize a NEF is using the mean function (cf. Theorem 3.6, page 74, of [Bro86]), $\mu_Q : \mathcal{U}_Q \to \mathbb{R}$, which is defined as

$$\mu_Q(u) = \int y\, Q_u(dy) = \frac{\int y \exp(uy) Q(dy)}{M_Q(u)}.$$

Since $Q_0 = Q$, clearly, $\mu_Q(0)$ is just the mean of $Q$. To minimize clutter, when $Q$ is clear from the context, we will write $\mu$ instead of $\mu_Q$. To illustrate the developments so far, we consider the example when the base distribution is an exponential distribution.

**Example 1** (Exponential distributions). *For $\lambda > 0$, let $Q$ be an exponential distribution with parameter $\lambda$: $Q(dy) = \mathbb{I}\{y \geq 0\} \lambda e^{-\lambda y} dy$. As is well known, the MGF of $Q$ takes the form $M_Q(u) = \frac{\lambda}{\lambda - u}$ when $u < \lambda$ and $M_Q(u) = \infty$ otherwise. Thus, $\mathcal{U}_Q = \{u \in \mathbb{R} : M_Q(u) < \infty\} = (-\infty, \lambda)$. The mean function takes the form of*

$$\mu(u) = \frac{\int_0^\infty \lambda y \exp(-\lambda y) \exp(uy) dy}{M_Q(u)} = \frac{1}{\lambda - u}, \qquad u < \lambda.$$

In what follows, we will need the following proposition to relate the central moments of $Q_u$ to the derivatives of $\psi_q$, the CGF.

**Proposition 2.** *Let $\mathcal{U}_Q^\circ$ be non-empty. Then, $\psi_Q$ is infinitely differentiable on $\mathcal{U}_Q^\circ$. Furthermore, the first three derivatives of $\psi_Q$ at $u \in \mathcal{U}_Q$ give the first moment, second and third central moments of $Q_u$.*

In the context of Example 1, Proposition 2 gives that $\psi_Q(u) = \log(\lambda) - \log(\lambda - u)$ when $u < \lambda$. Then, $\dot{\psi}_Q(u) = \frac{1}{\lambda - u}$, agreeing with our earlier computation.

## 3 Self-concordance of NEFs

This section contains the first set of main results of this paper. We start by giving our definition of self-concordance of NEFs, followed by a study of when this property is satisfied. We include a result that shows how self-concordance allows one to derive tail properties of members of the family from that of the base distribution.

In general, if the magnitude of a higher order derivative of a real function can be bounded in terms of a lower order derivative of the function, the function is said to be self-concordant [ST19]. This property is useful for studying how fast the function changes with its argument, as well as for deriving useful bounds on how well the function can be approximated by low order polynomials [NN94; Nes04; ST19]. In the context of single-parameter natural exponential families, we propose the following natural definition:

**Definition 1** (Self-concordant NEF). *Let $\mathcal{Q} = (Q_u)_{u \in \mathcal{U}}$ be a NEF with parameter set $\mathcal{U} \subset \mathcal{U}_Q^\circ$ and some base distribution $Q$. We say that $\mathcal{Q}$ is* self-concordant *if there exists a nonnegative valued function $\Gamma : \mathcal{U} \to \mathbb{R}_+$ such that*

$$|\ddot{\mu}(u)| \leq \Gamma(u)\dot{\mu}(u) \qquad \text{for all} \quad u \in \mathcal{U}. \tag{1}$$

*Any function $\Gamma : \mathcal{U} \to \mathbb{R}_+$ that satisfies Eq. (1) is called a* stretch function *of the NEF.*

This definition takes inspiration from the works of [Bac10; ST19; Fau+20] and [Jan+24] who define an analogous property (cf. Assumption 2 of [Jan+24]). By Proposition 2, we know that $\dot{\mu}(u)$ is the variance of $Q_u$, while $\ddot{\mu}(u)$ is the third central moment. Hence, $\dot{\mu}(u)$ is nonnegative, which explains why there is no absolute value on the right-hand side of Eq. (1).

According to our definition, we require that the (absolute value) of the second derivative of $\mu$ is bounded by the first derivative, up to the "stretch factor" $\Gamma(u)$. Clearly, provided that $\dot{\mu}$ is positive over $\mathcal{U}$, self-concordance is equivalent to stating that $\Gamma_Q(u) \doteq \frac{|\ddot{\mu}(u)|}{\dot{\mu}(u)}$ is finite valued over $\mathcal{U}$. When $\dot{\mu}(u) = 0$ for some $u \in \mathcal{U}$, one can show that $Q_u$ must be a Dirac distribution and hence so does $Q$ ($Q$ and $Q_u$ share their support). In this case, we also have $\ddot{\mu} \equiv 0$ and hence any nonnegative valued function is a valid stretch-function. In particular, $\Gamma_Q \equiv 0$ is also a valid stretch function.

It turns out that studying self-concordance property of distributions in detail can turn out much finer bounds than naively bounding $\ddot{\mu}$ and $\dot{\mu}$ separately.

**Example 3** (Avoiding exponential dependencies). *Consider a NEF $\mathcal{Q}$ with base distribution $Q$, a Bernoulli distribution with parameter $1/2$, we have $\mathcal{U}_Q = \mathbb{R}$, $\mu(u) = \frac{1}{1+e^{-u}}$ and thus $\dot{\mu} = \mu(1-\mu)$, $\ddot{\mu} = \dot{\mu}(1-2\mu)$. Hence $\mathcal{Q}$ is $\Gamma_Q$-self-concordant with $\Gamma_Q(u) \leq 1$ for all $u \in \mathbb{R}$. This is an example where naively bounding $\Gamma_Q$, by bounding the numerator and denominator separately over $[-s, s]$ for $s > 0$ gives a quantity of size $e^s$, which lags far behind the constant we obtained with a direct calculation, or what we can get from the result in Section 3.2, which show a scaling of order $O(s)$.*

As opposed to earlier literature where self-concordance is used [NN94; Bac10; ST19], we allow a non-constant stretch-factor $\Gamma$. As it turns out, this is necessary if $\mathcal{U} = \mathcal{U}_Q^\circ$ is to be allowed:

**Example 4** (Non-constant stretch factor). *Consider a NEF $\mathcal{Q}$ with base distribution $Q$. For $Q$ an exponential distribution with parameter $\lambda > 0$, $\mathcal{Q}$ is self-concordant over $\mathcal{U} = \mathcal{U}_Q = (-\infty, \lambda)$ with $\Gamma_Q(u) = \frac{2}{\lambda - u}$, $u < \lambda$. Indeed, a simple calculation gives that for $u < \lambda$, $\dot{\mu}(u) = (\lambda - u)^{-2}$ and $\ddot{\mu}(u) = 2(\lambda - u)^{-3}$, and so $|\ddot{\mu}(u)|/\dot{\mu}(u) = 2/(\lambda - u)$.*

As was shown above, for the NEF built on the exponential distribution with parameter $\lambda$, there is no constant stretch factor that makes the NEF self-concordant over the entirety of the natural parameter space. The main result of the next section shows that a non-constant stretch factor with growth similar

to the exponential always exists provided that the base distribution is *subexponential*.[2] As it turns out, the growth of the stretch factor plays an important role in applications: Among other things, it allows us to conclude that the tilted distributions are subexponential, as captured by the next result, which is a slightly generalized version of an analogous result of [Jan+24]:

**Lemma 5** (From self-concordance to light tails). *Let* $\mathcal{Q} = (Q_u)_{u \in \mathcal{U}}$ *be a NEF which is self-concordant with stretch function* $\Gamma : \mathcal{U} \to \mathbb{R}_+$ *where* $\mathcal{U}$ *is a subinterval of* $\mathcal{U}_Q^\circ = (a, b)$. *Then, for any* $u \in \mathcal{U}$,

$$\psi_{Q_u}(s) \leq s\mu(u) + s^2\dot{\mu}(u) \quad \text{for all} \quad s \in [-\log(2)/K, \log(2)/K] \cap (a - \inf\mathcal{U}, b - \sup\mathcal{U}), \quad (2)$$

*where* $K = \sup_{u \in \mathcal{U}} \Gamma(u)$.

Note that when $\mathcal{U}$ is a strict subset of $\mathcal{U}_Q^\circ$, $0 \in (a - \inf\mathcal{U}, b - \sup\mathcal{U})$ and thus the result is nontrivial as long as $K < \infty$. To interpret this result, recall that a centered distribution $Q$ is called *subexponential* with nonnegative parameters $(\nu, \alpha)$ if

$$\psi_Q(s) \leq s^2\nu^2/2, \qquad \text{for all } |s| \leq 1/\alpha$$

(cf. Definition 2.7 [Wai19]). A consequence of this is that the mean of $n$ independent random variables drawn from $Q$, with high probability, will be in an zero-centered interval of length $O(\sqrt{\nu^2/n} + \frac{1}{\alpha n})$. Here, the first term describes a "subgaussian" behavior, while the second a "pure subexponential" behavior. Assume for simplicity that $a = -b$ and $\inf\mathcal{U} = -\sup\mathcal{U}$. From Eq. (2) it follows that $Q_u$, when centered, is subexponential with parameters $\nu^2 = 2\dot{\mu}(u)$, twice the variance of $Q_u$, and $\alpha = \min(\log(2)/K, b - \sup\mathcal{U})$. In particular, we see that the growth of $\Gamma$ impacts the lower-order term in the confidence interval (the term $1/(\alpha n)$), but not the leading term, which is governed by the variance of $Q_u$. It follows that understanding how fast $\Gamma$ can grow as its argument approaches the boundary of $\mathcal{U}_Q$ is thus important, although its impact only appears in a low-order term.

We now turn to our first main result that shows that the growth of $\Gamma$ is at most inverse polynomial provided that the base distribution itself is subexponential.

## 3.1 Self-concordance with a subexponential base

We start with recalling equivalent definitions of subexponential distributions, which will be useful to interpret our results. In particular, according to Theorem 2.13 of [Wai19], given a zero-mean distribution $Q$ over the reals, the following are equivalent:

  (i) $Q$ is subexponential;

 (ii) The MGF of $Q$ is defined in an open neighborhood of zero;

(iii) For some positive constants $C, c > 0$, $\mathbb{P}(|Y| \geq t) \leq C\exp(-ct)$ where $Y \sim Q$.

It will be useful to have a quantitative version of this statement. For this, we separate the left and the right tails (as in some application they have different behaviors, which we may care about). The quantitative result for the right-tail is as follows (the result for the left-tail is omitted; it follows by symmetry):

**Proposition 6.** *Let* $c_1, C_1, c > 0$, $Y \sim Q$ *and assume that* $\mathbb{E}Y = 0$. *If*

$$\mathbb{P}(Y \geq t) \leq C_1\exp(-c_1 t) \qquad \text{for all } t \geq 0 \tag{3}$$

*then for any* $0 \leq \lambda < c_1$, $M_Q(\lambda) < 1 + \frac{C_1\lambda^2}{c_1(c_1 - \lambda)}$. *Furthermore, for any* $c > 0$ *such that* $M_Q(c) < \infty$, $\mathbb{P}(Y \geq t) \leq M_Q(c)e^{-ct}$ *holds for all* $t \geq 0$.

The first part is nontrivial; the second follows easily from Chernoff's method. The proof is given in Appendix B.

Let

$$\mathcal{E}_{\text{right}}(c_1, C_1) = \{Q : Y = X - \mathbb{E}X \text{ satisfies Eq. (3) where } X \sim Q\}.$$

---

[2]Here, we follows the terminology used in the concentration of measure literature where these distributions are also knowns as subgamma distributions [BLM13; Ver18; Wai19], or light-tailed distributions. This is to be contrasted to the use of the same term in the theory of heavy-tailed distributions, where subexponential refers to a much larger class of distributions [GK98].

In words, $\mathcal{E}_{\text{right}}$ is the class of distributions over the reals whose right-tail displays an exponential decay governed with the rate parameter $c_1 > 0$ and scaling constant $C_1 > 0$. Similarly, we let $\mathcal{E}_{\text{left}}(c_1, C_1)$ be the class of distributions $Q$ over the reals such that for $X \sim Q$, $Y = \mathbb{E}X - X$ satisfies Eq. (3). With this notation, the first part of the previous proposition is equivalent to that if $Q \in \mathcal{E}_{\text{right}}(c_1, C_1)$ is centered then $M_Q$ stays below the function $\lambda \mapsto 1 + \frac{C_1 \lambda^2}{c_1(c_1 - \lambda)}$ over the interval $[0, c_1)$. In particular, this means that $\mathcal{U}_Q$ contains $[0, c_1)$.

With this, we are ready to present our theorem that establishes the self-concordance property of NEFs with subexponential tails.

**Theorem 7.** *Let $Q \in \mathcal{E}_{right}(c_1, C_1) \cap \mathcal{E}_{left}(c_2, C_2)$ for some positive constants $c_i, C_i$, $i = 1, 2$. Then, the NEF $\mathcal{Q} = (Q_u)_{u \in \mathcal{U}}$ is self-concordant. Moreover, the function $\Gamma : \mathcal{U} \to \mathbb{R}_+$ defined by*

$$
\Gamma(u) = \begin{cases} \frac{3}{2}\left[2eC_1 c_1\left(\frac{1}{c_1 - u}\right)^2 + \frac{ub}{c_1 - u}\right] + G_Q(C_1, C_2, c_1, c_2) & \text{if } 0 \le u < c_1 \\ \frac{3}{2}\left[2ec_2 C_2\left(\frac{1}{c_2 + u}\right)^2 + \frac{-ub}{c_2 + u}\right] + G_Q(C_2, C_1, c_2, c_1) & \text{if } -c_2 < u < 0, \end{cases}
$$

*is a stretch function for the NEF $\mathcal{Q}$, where $G_Q(M_1, M_2, m_1, m_2)$ is a polynomial whose coefficients depend on $Q$.*

The exact expression of $G_Q$ can be found in Eq. (8). Let $\mathcal{Q}$ be a NEF with base distribution $Q$ when $Q$ is a zero-mean Laplace distribution with variance $2\lambda^2$. In this case, we can choose $c_1 = c_2 = 1/\lambda$, $C_1 = C_2 = 1/2$. The actual stretch function is $\Gamma_Q(u) = \frac{8\lambda^2|u|}{(1/\lambda - u)(1/\lambda + u)(3\lambda^4 u^2 + \lambda^2)}$. Thus, the above theorem gives the correct behavior in that both the stretch function from the theorem and the actual stretch function $\Gamma_Q$ blow up with the inverse of the distance to the boundaries of $\mathcal{U}_Q$, except that the actual growth scales linearly with the inverse distance, while the theorem gives a quadratic scaling. It remains an open problem to see whether this quadratic order can be improved.

The following corollary is an immediate result of Lemma 5 and Theorem 7 (Appendix C.3), but can also be proved directly from the definitions (and we include a direct proof as part of the proof of Theorem 7).

**Corollary 8** (Distributions in regular NEFs are subexponential)**.** *Let $u \in \mathcal{U}_Q^\circ$. Then $Q_u$ is subexponential both on left and right.*

Because of this result, there is essentially no loss in generality in only considering the subexponential case when working with NEFs. In particular, self-concordance in NEFs is "almost free".

**Proof sketch for Theorem 7** We sketch here the result for $\mu(0) = 0$, $\dot{\mu}(0) > 0$ and $u \ge 0$. The arguments used to extend the result to all cases can be found in Appendix C. By Proposition 2, bounding the stretch function of a NEF amounts to showing

$$
\Gamma_Q(u) := \frac{\int |(y - \mu(u))|^3 Q_u(dy)}{\int (y - \mu(u))^2 Q_u(dy)} \le \Gamma(u),
$$

where the division by $\text{Var}(Q_u) = \int (y - \mu(u))^2 Q(dy)$ is justified by Lemma 12. We split the proof of that upper bound into two steps: controlling the variance and the absolute third moment.

**Step 1: Controlling the variance** Since $\dot{\mu}(0) = \text{Var}(Q) > 0$, there exists a $Q$-dependent constant $\eta > 0$ and an interval $[-b, -a] \subset \mathbb{R}_{<0}$ s.t. $Q([-b, -a]) \ge \eta$ (Lemma 16). With this observation, we can show (Lemma 17):

$$
\dot{\mu}(u) \ge a^2 \eta \frac{e^{-ub}}{M_Q(u)}. \tag{4}
$$

Thus, the second moment decreases at most exponentially with the parameter $u$.

**Step 2: Controlling the absolute third central moment** First, we use a classical result on moments of random variables (see the proof of $i \Rightarrow ii$ for prop.2.5.2 in [Ver18]):

$$
\int |y - \mu(u)|^3 Q_u(dy) = \int_0^B 3t^2 \mathbb{P}(|Y - \mu(u)| \ge t)dt + \int_B^\infty 3t^2 \mathbb{P}(|Y - \mu(u)| \ge t)dt
$$

$$
\le \frac{3}{2}B\dot{\mu}(u) + \int_B^\infty 3t^2 \mathbb{P}(|Y - \mu(u)| \ge t)dt, \tag{5}
$$

where $Y \sim Q_u$ and $B$ is a constant to be optimized. It should remain small enough for the first term not to blow up. On the other hand, it should be large enough for the second term divided by the lower bound obtained on $\dot{\mu}(u)$ to remain controlled.

To upper bound the second term in Eq. (5), we start by showing that the right tail of the tilted distribution $Q_u$ is also subexponential (Lemma 18):

$$Q_u((t, \infty)) \lesssim \frac{1}{M_Q(u)} e^{-(c_1-u)t} \frac{1}{c_1 - u} \quad \text{for} \quad 0 \le u < c_1. \tag{6}$$

Following this lemma, we get an upper bound on $\mu(u)$ (Lemma 20):

$$0 \le \mu(u) \lesssim \left( \frac{1}{c_1 - u} \right)^2 \quad \text{for} \quad 0 \le u < c_1. \tag{7}$$

In the proof, we bound separately the positive and negative values in the second term of Eq. (5):

$$\int_B^\infty 3t^2 \mathbb{P}(|Y - \mu(u)| \ge t) dt = \underbrace{\int_B^\infty 3t^2 \mathbb{P}(Y \ge \mu(u) + t) dt}_{\spadesuit} + \underbrace{\int_B^\infty 3t^2 \mathbb{P}(Y \le \mu(u) - t) dt}_{\heartsuit}.$$

We give here the sketch of proof on the bound of $\spadesuit$ as the proof for bounding $\heartsuit$ is nearly identical. From plugging in Eq. (6), we get:

$$\spadesuit \lesssim \frac{1}{M_Q(u)} \frac{1}{c_1 - u} \int_B^\infty 3t^2 e^{-(c_1-u)(t+\mu(u))} dt.$$

By choosing $B \gtrsim \frac{ub}{c_1 - u} + \left( \frac{1}{c_1 - u} \right)^2$, some algebra gives:

$$\spadesuit \lesssim \frac{e^{-ub}}{M_Q(u)} \left( \frac{1 + u^2 b^2}{c_1^3 C_1^3} \right).$$

Setting $B \gtrsim \left( \frac{1}{c_1 - u} \right)^2 + \frac{ub}{c_2 + u}$ gives a similar bound on $\heartsuit$. Chaining the bounds on $\spadesuit$ and $\heartsuit$ with Eq. (5) and Lemma 17 finishes the proof. $\qquad\square$

## 3.2 Self-concordance with a subgaussian base

In this section we refine the previous result for NEFs by considering the case when the base distribution is subgaussian. Let $\sigma > 0$. Recall that a centered distribution $Q$ is $\sigma$-subgaussian if for all $u \in \mathbb{R}$, $M_Q(u) \le e^{\sigma^2 u^2 / 2}$ (or, equivalently, $\psi_Q(u) \le \sigma^2 u^2 / 2$ for any $u \in \mathbb{R}$). A non-centered distribution is $\sigma$-subgaussian, if it is subgaussian after centering. Similarly to the subexponential case, one can show that a centered distribution $Q$ is subgaussian if and only if for some $\tau, C > 0$, it holds that for any $t \ge 0$, $\mathbb{P}(|X| \ge t) \le C \exp(-t^2 / (2\tau^2))$ where $X \sim Q$ (cf. Proposition 2.5.2 of [Ver18]).[3]

Our promised result is as follows:

**Theorem 9.** *Let $Q$ be subgaussian. Then, the NEF $\mathcal{Q} = (Q_u)_{u \in \mathbb{R}}$ is self-concordant and $\Gamma_Q(u) = O(|u|)$, $u \in \mathbb{R}$.*

As it turns out, the linear growth exhibited in the previous result is tight for NEFs with a subgaussian base distribution:

**Theorem 10.** *There exists a distribution $Q$ that is subgaussian such that $\limsup_{u \to \infty} \Gamma_Q(u)/u > 0$.*

Again, this shows that even if we stay with subgaussian distributions, it would be limiting to only consider NEFs that are self-concordant with a bounded (or constant) stretch function over their natural parameter space.

---

[3]As for the quantitative relation between the parameters, it can be shown ([RH23]) that if for all $u \in \mathbb{R}$, $M_Q(u) \le e^{\sigma^2 u^2 / 2}$, then $\mathbb{P}(|X| \ge t) \le 2 \exp(-t^2 / (2\sigma^2))$, and if $\mathbb{P}(|X| \ge t) \le 2 \exp(-t^2 / (2\sigma^2))$, then for all $u \in \mathbb{R}$, $M_Q(u) \le e^{4\sigma^2 u^2}$.

# 4 Generalized Linear Bandits

In this section we apply the self-concordance property of subexponential NEFs in order to derive novel confidence sets and regret bounds for subexponential generalized linear bandits. To our knowledge, these are the first such results for parametric bandits with subexponential rewards.

## 4.1 Bandit model

Following Filippi, Cappe, Garivier, and Szepesvári [Fil+10], we consider stochastic generalized linear bandit (GLB) models $\mathcal{G}$ specified by a tuple $(\mathcal{X}, \Theta, \mathcal{Q})$, where $\mathcal{X} \subseteq \mathbb{R}^d$ is a non-empty arm set, $\Theta \subseteq \mathbb{R}^d$ is a non-empty set of potential parameters, both closed for convenience, $\mathcal{Q} = (Q_u)_{u \in \mathcal{U}_Q}$ is a NEF with base distribution $Q$. Without the loss of generality we assume that $\mathcal{X}$ is a compact subset of the Euclidean unit ball of $\mathbb{R}^d$.

In each round $t \in \mathbb{N}^+$, the learner selects and plays an arm $X_t \in \mathcal{X}$. As a response, they receive a reward $Y_t$ sampled from the distribution $Q_{X_t^\top \theta_\star}$, where $\theta_\star \in \mathbb{R}^d$ is a parameter of the bandit environment, which is initially unknown to the learner. The learner's goal is to maximize its total expected reward. The GLB is *well-posed* when

$$\mathcal{U} \doteq \{x^\top \theta : x \in \mathcal{X}, \theta \in \Theta\} \subseteq \mathcal{U}_Q^\circ$$

holds, which we assume from now on. The condition that $\mathcal{U} \subseteq \mathcal{U}_Q$ simply ensures that the reward distributions $Q_{x^\top \theta}$ are defined regardless of the value of $(x, \theta) \in \mathcal{X} \times \Theta$. We require the stronger condition $\mathcal{U} \subseteq \mathcal{U}_Q^\circ$ to exclude the boundaries of the interval $\mathcal{U}_Q$. This way we avoid pathologies that arise when a parameter reaches the boundary of $\mathcal{U}_Q$ (e.g., when $Q$ is the exponential distribution with parameter $\lambda$, the mean and variance of $Q_u$ grow unbounded as $u$ approaches $\lambda$ from below).

The expected reward in round $t$ given that the learner plays $X_t$ is $\mathbb{E}[Y_t|X_t] = \mu(X_t^\top \theta_\star)$. The performance of the learner will be assessed by their pseudo regret $R(T)$, which is the total cumulative shortfall of the mean reward of the arms the learner chose relative to optimal choice:

$$R(T) = \sum_{t=1}^{T} \mu(x_\star^\top \theta_\star) - \mu(X_t^\top \theta_\star).$$

Here, $x_\star \in \arg\max_{x \in \mathcal{X}} \mu(x^\top \theta_\star)$ is the arm that results in the best possible expected reward in a round. For simplicity, we assume that such an arm exists. We establish guarantees of our algorithm for a subclass of GLBs, captured by the following assumption:

**Assumption 1** (Subexponential base)**.** *The base distribution $Q$ is subexponential both on the left and the right. In particular, $Q \in \mathcal{E}_{right}(c_1, C_1) \cap \mathcal{E}_{left}(c_2, C_2)$ for some $c_i, C_i$ $(i = 1, 2)$ positive numbers. Furthermore, $-c_2 < \inf \mathcal{U} \leq \sup \mathcal{U} < c_1$ and $c_1, c_2$ are known to the learner.*

Note that in a well-posed GLB, $\inf \mathcal{U}_Q < \inf \mathcal{U} \leq \sup \mathcal{U} < \sup \mathcal{U}_Q$. In light of this, the assumption just stated boils down to whether $0 \in \mathcal{U}_Q^\circ$, which, as discussed, is free when $\mathcal{U}_Q^\circ \neq \emptyset$. Indeed, when $0 \in \mathcal{U}_Q^\circ$ holds, $\inf \mathcal{U}_Q < 0 < \sup \mathcal{U}_Q$ and one can always find positive values $c_1, c_2$ such that $\inf \mathcal{U}_Q < -c_2 < \inf \mathcal{U} \leq \sup \mathcal{U} < c_1 < \sup \mathcal{U}_Q$. Then, from Proposition 6 and some extra calculation one can conclude that the $Q \in \mathcal{E}_{right}(c_1, e^{-c_1 \mathbb{E}X} M_Q(c_1)) \cap \mathcal{E}_{left}(c_2, e^{-c_2 \mathbb{E}X} M_Q(-c_2))$. Since it is assumed that the learner has access to $Q$, we see that Assumption 1 can be satisfied whenever $0 \in \mathcal{U}_Q^\circ$, which we think is a rather mild assumption. We will also for the sake of simplicity assume that the learner has access to $S_0 = \sup\{\|\theta\| : \theta \in \Theta\}$, $S_2 = \inf \mathcal{U}$, $S_1 = \sup \mathcal{U}$. These values will be used in setting the parameters of the algorithm. Note that it is not critical that the learner knows these exact values; appropriate bounds suffice. We also assume that the learner is given access to an upper bound on the worst-case variance over the parameter space $\mathcal{U}$:

**Assumption 2** (Bounded Variance)**.** *The learner is given $L \geq 1$ such that $\sup_{u \in \mathcal{U}} \dot{\mu}(u) \leq L$.*

Note that since the GLB is well-posed, $\sup_{u \in \mathcal{U}} \dot{\mu}(u) < \infty$ is automatically satisfied. Also, there is no loss of generality in assuming $L \geq 1$. A crude upper bound on $\sup_{u \in \mathcal{U}} \dot{\mu}(u)$ is $C_1 e/(c_1 - S_1)^3 \vee C_2 e/(c_2 + S_2)^3$, so this assumption is implied by the previous one.

## 4.2 The OFU-GLB algorithm and its regret

Just like the previous works [Fil+10; LLZ17; Fau+20] which considered special cases of the generalized linear bandit problem, our algorithm follows the "optimism in the face of uncertainty" principle. In each time step, the algorithm constructs a confidence set $\mathcal{C}_t$, based on past information, that contains the unknown parameter $\theta_\star$ with a controlled probability. Next, the algorithm chooses a parameter $\theta_t$, in the confidence set $\mathcal{C}_t$, and an underlying action $X_t \in \mathcal{X}$ such that the mean reward underlying $X_t$ and $\theta_t$ is as large as plausibly possible. Since $\mu = \mu_Q$ is guaranteed to be an increasing function (recall that $\dot\mu(u)$ is the variance of $Q_u$ and is hence nonnegative), it suffices to find the maximizer of $x^\top\theta$ where $(x, \theta) \in \mathcal{X} \times \Theta$. We call our algorithm, shown in Algorithm 1, OFU-GLB (Optimism in the Face of Uncertainty in Generalized Linear Bandits). The main novelty here is that our bandit model makes minimal assumptions.

---

**Algorithm 1** The OFU-GLB Algorithm

---

**Require:** GLB instance $\mathcal{G} = (\mathcal{X}, \Theta, \mathcal{Q})$
    **for** $t = 1, 2, \ldots$ **do**
        Construct $\mathcal{C}_t \subset \Theta$ based on $((X_s, Y_s))_{s<t}$ and $\mathcal{G}$
        Compute $(X_t, \theta_t) \in \arg\max_{(x,\theta)\in\mathcal{X}\times\mathcal{C}_t} x^\top\theta$
        Select arm $X_t$ and receive reward $Y_t \sim Q_{X_t^\top\theta_\star}$
    **end for**

---

The confidence set is based on ideas from the work of Janz, Liu, Ayoub, and Szepesvári [Jan+24]. Note that this paper analyzed a randomized method for those GLBs where $\mathcal{U}_Q = \mathbb{R}$ and $\sup_{u\in\mathcal{U}} \Gamma_Q(u) < \infty$. The assumption that $\mathcal{U}_Q = \mathbb{R}$ is restrictive, as it does not allow many common distributions (e.g., the exponential distribution). Thanks to Theorem 7, under our assumptions, $\sup_{u\in\mathcal{U}} \Gamma_Q(u) < \infty$ follows. Then, an appropriate confidence set can be constructed based on Lemma 5, which also extended the corresponding result of Janz, Liu, Ayoub, and Szepesvári [Jan+24]. While the confidence set construction is based on the ideas of Janz, Liu, Ayoub, and Szepesvári [Jan+24], the main steps of the analysis are taken from [AFC21] who analyzed logistic bandits, which are 1-self-concordant. Our result follows by carefully modifying the proof of [Jan+24] and carefully propagating both the effect of replacing their confidence set with a different one, and the effect of $\sup_{u\in\mathcal{U}} \Gamma_Q(u) > 1$. This leads to the main result on GLBs:

**Theorem 11** (Regret upper bound of OFU-GLB). *Let $\delta \in (0, 1]$ and $T$ a positive integer and consider a well-posed GLB model $\mathcal{G} = (\mathcal{X}, \Theta, \mathcal{Q})$ and assume that Assumptions 1 and 2 hold. For $\theta_\star \in \Theta$, let $\kappa(\theta_\star) = \frac{1}{\dot\mu(x_\star^\top\theta_\star)}$ and let $\mathrm{Regret}(T, \theta_\star)$ stand for the $T$-round regret of OFU-GLB when it interacts with a GLB specified by $\theta_\star$. Then, with an appropriate construction of $\mathcal{C}_t$, for any $\theta_\star \in \Theta$, it holds that with probability at least $1 - \delta$,*

$$\mathrm{Regret}(T) = \tilde{\mathcal{O}}\left(d\sqrt{\dot\mu(x_\star^\top\theta_\star)T} + d\kappa(\theta_\star)\right),$$

*where $\tilde{O}(\cdot)$ hides polylogarithmic factors in $T, d, L, 1/\delta$ and constants that depend on the base distribution Q.*

This result extends the class of distributions for which OFU algorithms with parametric models achieve sublinear regret. Previous results in the literature [Fil+10; APS11; LLZ17; Fau+20; Jun+21; Jan+24] all assume that the base distribution is a natural exponential family with subgaussian tail and prove that the OFU algorithm enjoys sublinear regret. Thus, Theorem 11 extends the class of distributions for which optimistic algorithms enjoy sublinear regret to any natural exponential family with subexponential base distribution.

An essential quality of the result is that it makes the dependence of the regret on the instance $\theta_\star$ explicit. Recalling that in a NEF, $\dot\mu(u)$ is the variance of the tilted distribution $Q_u$, we see that the leading term (shown as the first term on the right-hand side of the last display) scales with the variance of the optimal arm's reward distribution. In queuing theory, the service times of agents (actions) in an environment are often modeled as exponentially distributed random variables [GN67]. When aiming to minimize service times (maximize negative reward) with an exponential bandit model (with mean function $\mu(x) = -1/x$), the variance of the optimal arm's service time lower bounds that of the other arms. Furthermore, the dependence on $\kappa$, a term that is inversely proportional to the optimal variance,

is pushed to a second order term. In logistic bandits, $\kappa$ can be exponentially large in the size of the parameter set $S_0$ and thus much attention has been focused on mitigating its effect [Fau+20; Jun+21; Jan+24]. Our regret bound also matches the lower bound in logistic bandits given by [AFC21], thus our analysis is tight for this special case.

## 5    Conclusions and Future Work

The main contribution of this work establishes that all subexponential NEFs are self-concordant with a polynomial-sized stretch factor. We then applied this finding and derived regret bounds for subexponential GLBs that scale with the variance of the optimal arm's reward, which is the smallest variance amongst all arm's rewards in problems such as: minimizing service times [GN67] or minimizing insurance claim severity (dollars lost per claim) [GKT16].

Our findings also have implications when performing maximum likelihood estimation with subexponential NEFs, which includes a rich family of generalized linear models (GLMs). Since the log loss in a NEF is the sum of a linear function and the NEF's CGF, the GLM's loss is self-concordant in the sense of (say) [Bac14] whenever the NEF is self-concordant. While this is outside of the scope of our paper, it follows that this family of GLMs enjoy: $(i)$ fast rates of convergence to the minimizer for regularized empirical risk minimization [Mar+19], $(ii)$ fast rates for averaged stochastic gradient descent [Bac14] and $(iii)$ fast rates for constrained optimization with first-order methods [Dvu+20], without restrictive conditions on bounded responses, which previous works had to assume to achieve these results.

One interesting direction for future work would be either deriving a matching lower bound on the stretch function for subexponential NEFs or tighter analysis that matches the lower bound for subgaussian NEFs. Another potential avenue for future work would be in extending our results to other exponential families, beyond NEFs.

## Acknowledgments and Disclosure of Funding

The authors would like to thank Iosif Pinelis for his helpful insights to our work and Samuel Robertson for reviewing and earlier version of this manuscript. Shuai Liu would like to acknowledge Alireza Bakhtiari for helpful discussions. Csaba Szepesvári also gratefully acknowledges funding from the Canada CIFAR AI Chairs Program, Amii and NSERC.

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

## A  Extra notation

The following extra notation will be used in the appendix: For vector $x \in \mathbb{R}^d$, we let $\|x\|$ denote its $\ell_2$-norm and for positive definite matrix $M \in \mathbb{R}^{d \times d}$, we use $\|x\|_M = \sqrt{x^\top M x}$ to denote its $M$-weighted $\ell_2$-norm.

## B  On subexponential (or "light tailed") distributions

We first prove Proposition 6, which we repeat for the convenience of the reader:

**Proposition 6.** *Let $c_1, C_1, c > 0$, $Y \sim Q$ and assume that $\mathbb{E}Y = 0$. If*

$$\mathbb{P}(Y \geq t) \leq C_1 \exp(-c_1 t) \qquad \text{for all } t \geq 0 \tag{3}$$

*then for any $0 \leq \lambda < c_1$, $M_Q(\lambda) < 1 + \frac{C_1 \lambda^2}{c_1(c_1 - \lambda)}$. Furthermore, for any $c > 0$ such that $M_Q(c) < \infty$, $\mathbb{P}(Y \geq t) \leq M_Q(c)e^{-ct}$ holds for all $t \geq 0$.*

We follow the proof of Theorem 2.13 from the book of Wainwright [Wai19].

*Proof.* We start with the second part. For this let $t \geq 0$, $c > 0$. Then, by Chernoff's method, we have

$$\mathbb{P}(Y \geq t) \leq \mathbb{E}[e^{cX}]e^{-ct} = M_Q(c)e^{-ct}.$$

The first part requires more work. Let us start by bounding the $p$-th moment of the positive part of $Y$, which we denote by $Y_+$ (hence, $Y_+ = \max(Y, 0)$). We have

$$
\begin{aligned}
\mathbb{E}[Y_+^p] &= \int_0^\infty \mathbb{P}(Y_+^p \geq u)du \\
&= p \int_0^\infty \mathbb{P}(Y_+ \geq t)t^{p-1}dt && \text{(change of variables with } u = t^p) \\
&= p \int_0^\infty \mathbb{P}(Y \geq t)t^{p-1}dt && \text{(for } t > 0, \{Y_+ \geq t\} = \{Y \geq t\}) \\
&\leq C_1 p \int_0^\infty e^{-c_1 t}t^{p-1}dt && \text{(assumption on } Y) \\
&\leq \frac{C_1 p}{c_1^p} \int_0^\infty e^{-u}u^{p-1}du && \text{(change of variables with } u = c_1 t) \\
&= \frac{C_1 p}{c_1^p}\Gamma(p-1) && \text{(definition of the } \Gamma \text{ function)} \\
&= \frac{C_1}{c_1^p}p! && \text{(property of the } \Gamma \text{ function)}
\end{aligned}
$$

Now let $0 \leq \lambda < c_1$. Since $Y \leq Y_+$, we have $M_Q(\lambda) = \mathbb{E}[e^{\lambda Y}] \leq \mathbb{E}[e^{\lambda Y_+}]$. Hence, by the power-series expansion of the exponential, we get

$$
\begin{aligned}
M_Q(\lambda) &\leq \mathbb{E}\left[e^{\lambda Y_+}\right] \\
&= 1 + \sum_{p=2}^\infty \lambda^p \frac{\mathbb{E}\left[Y_+^p\right]}{p!} \\
&\leq 1 + C_1 \sum_{p=2}^\infty \left(\frac{\lambda}{c_1}\right)^p \\
&\leq 1 + C_1 \frac{\lambda}{c_1}\frac{\lambda}{c_1 - \lambda}. \qquad \square
\end{aligned}
$$

We note in passing that since $\lambda < c_1$, $1 + C_1 \frac{\lambda}{c_1}\frac{\lambda}{c_1 - \lambda} \leq 1 + C_1 \frac{\lambda}{c_1 - \lambda}$. That Eq. (3) implies that $M_Q(\lambda) \leq 1 + C_1 \frac{\lambda}{c_1 - \lambda}$ can also be obtained by refining the proof of Theorem 2.13 from the book of Wainwright [Wai19].

## C  Proof of Theorem 7

For the convenience of the reader we restate the theorem to be proven:

**Theorem 7.** *Let $Q \in \mathcal{E}_{right}(c_1, C_1) \cap \mathcal{E}_{left}(c_2, C_2)$ for some positive constants $c_i, C_i$, $i = 1, 2$. Then, the NEF $\mathcal{Q} = (Q_u)_{u \in \mathcal{U}}$ is self-concordant. Moreover, the function $\Gamma : \mathcal{U} \to \mathbb{R}_+$ defined by*

$$\Gamma(u) = \begin{cases} \frac{3}{2}\left[2eC_1c_1\left(\frac{1}{c_1-u}\right)^2 + \frac{ub}{c_1-u}\right] + G_Q(C_1, C_2, c_1, c_2) & \text{if } 0 \leq u < c_1 \\ \frac{3}{2}\left[2ec_2C_2\left(\frac{1}{c_2+u}\right)^2 + \frac{-ub}{c_2+u}\right] + G_Q(C_2, C_1, c_2, c_1) & \text{if } -c_2 < u < 0, \end{cases}$$

*is a stretch function for the NEF $\mathcal{Q}$, where $G_Q(M_1, M_2, m_1, m_2)$ is a polynomial whose coefficients depend on $Q$.*

Note that the function $\Gamma$ as defined above is non-decreasing on $\mathcal{U} \cap \mathbb{R}^+$ and non-increasing on $\mathcal{U} \cap \mathbb{R}^-$.

The actual form of $G_Q$ is as follows: let $a, b, \eta > 0$ be such that $Q([-b + \mu(0), -a + \mu(0)]) > \eta$ and $-a < 0$. Then,

$$G_Q(M_1, M_2, m_1, m_2) = \frac{3}{2}b + \frac{1}{a^2\eta}\left(\frac{204}{e^3 m_1^3 M_1^3} + \frac{6b^2}{e^3 m_1 M^3} + \frac{81M_2 + 9M_2 m_1^2 b^2}{m_2^3}\right) \quad (8)$$

We note in passing that these values are not controlled by the tail behavior of the base distribution $Q$. This can be seen, for example, by considering $Q(dy) = (1 - \eta)\mathbb{I}(y \geq 0)e^{-y}dy + \eta e^{-b}\delta_{\{-b\}}(dy)$. Tedious calculation shows that $\lim_{u \to -\infty} \Gamma_Q(u) = \Omega(b)$ as $b \to \infty$. And because $Q \in \mathcal{E}_{right}(1, 1) \cap \mathcal{E}_{left}(1, 1)$, this shows that the tail behavior of $Q$ is indeed insufficient to control the behavior of $\Gamma_Q$.

We will prove this result in three parts: *(i)* $\mathrm{Var}(Q) = 0$ *(ii)* $\mathrm{Var}(Q) > 0$ and $\mathcal{U} = [0, c_1)$, *(iii)* $\mathrm{Var}(Q) > 0$ and $\mathcal{U} = (-c_2, 0]$. The result follows from combining these cases.

The main work is to prove the result for $\mathcal{U} = [0, c_1)$, which is done in Proposition 13. Case *(iii)* is handled by "reflection around the origin" (Corollary 14). Case *(i)* is handled in Lemma 12 by showing that $\Gamma_Q \equiv 0$ if $\mathrm{Var}(Q) = 0$.

We start with case *(i)*, the degenerate case when the variance of $Q$ is zero.

**Lemma 12.** *If $\mathrm{Var}(Q) = 0$ then $\mathcal{U}_Q = \mathbb{R}$, $Q_u = Q$ for all $u \in \mathbb{R}$, and $\Gamma_Q \equiv 0$. If $\mathrm{Var}(Q) > 0$ then $\dot{\mu}$ is strictly positive over the entire set $\mathcal{U}_Q^\circ$.*

*Proof.* If $\mathrm{Var}(Q) = 0$, then $Q$ is a Dirac on some $\{v\}$. Then for all $u \in \mathbb{R}$, $M_Q(u) = \exp(uv) < \infty$ hence $\psi_Q = \log M_Q$ is supported on $\mathbb{R}$ and

$$Q_u(A) = \begin{cases} \frac{1}{M_Q(u)}\exp(uv) = 1 & \text{if } v \in A \\ 0 & \text{otherwise.} \end{cases}$$

Hence $Q_u = Q$ and $\mathcal{Q} = (Q_u)_{u \in \mathbb{R}}$ is trivially self-concordant with the stretch function defined to be $\Gamma_Q \equiv 0$.

For the second part, by Proposition 2, we have that $\dot{\mu}(u) = \int(y - \mu(u))^2 Q_u(dy) \geq 0$. We will show $\dot{\mu}(u) \neq 0$ by contradiction. Assume there exists $u_0 \in \mathcal{U}_Q^\circ$ such that $\dot{\mu}(u_0) = \int(y - \mu(u_0))^2 Q_{u_0}(dy) = 0$, then it follows that $Q_{u_0}(dy)$ is a Dirac on $\{\mu(u_0)\}$, which implies for all $A \in \mathcal{B}(\mathbb{R})$

$$Q(A) = M_Q(u_0) \cdot Q_{u_0}(A) = \begin{cases} M_Q(u_0) & \{\mu(u_0)\} \not\subseteq A \\ 0 & \text{otherwise} \end{cases}$$

where $\mathcal{B}(\mathbb{R})$ denotes the Borel sets on $\mathbb{R}$. Then it follows that $Q(dy)$ is also a Dirac on $\mathbb{R}$, which contradicts that $\mathrm{Var}(Q) > 0$. $\square$

Consider now the case when $\mathrm{Var}(Q) > 0$. By the result just stated $\dot{\mu}$ is bounded away from zero over $\mathcal{U}_Q^\circ$ and hence it is safe then to define $\Gamma_Q$ with the ratio $\frac{|\ddot{\mu}(u)|}{\dot{\mu}(u)}$:

$$\Gamma_Q(u) = \frac{|\ddot{\mu}(u)|}{\dot{\mu}(u)}, \qquad u \in \mathcal{U}_Q^\circ.$$

Thus, in order to show our result, it suffices to show $\Gamma_Q \leq \Gamma$ with the function $\Gamma$ as stated in the theorem. Thus, we will study $\Gamma_Q$. First, notice that for all $u \in \mathcal{U}_Q^\circ$, by Proposition 2,

$$\Gamma_Q(u) = \frac{|\int (y - \mu(u))^3 Q_u(dy)|}{\int (y - \mu(u))^2 Q_u(dy)} \leq \frac{\int |y - \mu(u)|^3 Q_u(dy)}{\int |y - \mu(u)|^2 Q_u(dy)}.$$

Let us now state the results that are concerned with cases *(ii)* and *(iii)* mentionned above. For case *(ii)*, i.e., when $\mathcal{U} = [0, c_1)$ we have the following result:

**Proposition 13.** *Let* $Q \in \mathcal{E}_{right}(c_1, C_1) \cap \mathcal{E}_{left}(c_2, C_2)$ *and* $\mathcal{U} = [0, c_1)$. *Define* $\Gamma : \mathcal{U} \to \mathbb{R}_+$ *by*

$$\Gamma(u) = \frac{3}{2}\left[2c_0\left(\frac{1}{c_1 - u}\right)^2 + \frac{ub}{c_1 - u} + \frac{|u|b}{c_1 + |u|}\right] + \frac{1}{a^2\eta}\left(\frac{204 + 6u^2b^2}{c_0^3} + \frac{81C_2 + 9C_2u^2b^2}{(u + c_2)^3}\right)$$

$$\leq \frac{3}{2}\left[2c_0\left(\frac{1}{c_1 - u}\right)^2 + \frac{ub}{c_1 - u}\right] + G_Q(C_1, C_2, c_1, c_2),$$

*where* $c_0 = C_1 \cdot c_1 \cdot e$. *Then, for appropriate values of* $\eta, a, b > 0$ *that depend on the base distribution* $Q$, *we have* $\Gamma_Q \leq \Gamma$ *over* $\mathcal{U}$.

For case *(iii)*, i.e., when $\mathcal{U} = (-c_2, 0]$, we have the following result:

**Corollary 14.** *Let* $Q \in \mathcal{E}_{right}(c_1, C_1) \cap \mathcal{E}_{left}(c_2, C_2)$ *and* $\mathcal{U} = (-c_2, 0]$. *Define* $\Gamma : \mathcal{U} \to \mathbb{R}_+$ *by*

$$\Gamma(u) = \frac{3}{2}\left[2c_0\left(\frac{1}{c_2 - |u|}\right)^2 + \frac{|u|b}{c_2 - |u|} + \frac{|u|b}{c_1 + |u|}\right] + \frac{1}{a^2\eta}\left(\frac{204 + 6u^2b^2}{c_0^3} + \frac{81C_1 + 9C_1u^2b^2}{(|u| + c_1)^3}\right)$$

$$\leq \frac{3}{2}\left[2c_0\left(\frac{1}{c_2 - |u|}\right)^2 + \frac{|u|b}{c_2 - |u|}\right] + G_Q(C_2, C_1, c_2, c_1),$$

*where* $c_0 = C_2 \cdot c_2 \cdot e$. *Then, for appropriate values of* $\eta, a, b > 0$ *that depend on the base distribution* $Q$, *we have* $\Gamma_Q \leq \Gamma$ *over* $\mathcal{U}$.

### C.1   Proof of Proposition 13

The key idea is to convert $\ddot{\mu}(\cdot)$ and $\dot{\mu}(\cdot)$ to third and second central moments respectively by Proposition 2 and bound the third central moment in terms of the second central moment (variance).

We start by an elementary observation that says that the self-concordance properties of a NEF do not change when the base distribution is shifted by a constant:

**Lemma 15.** *Let* $Y \sim Q$, $c \in \mathbb{R}$ *and define* $Q^{+c}$ *to be the distribution of* $Y + c$. *Then,* $\mathcal{U}_Q = \mathcal{U}_{Q^{+c}}$, $M_{Q+c}(u) = e^{-uc}M_Q(u)$ *for all* $u \in \mathbb{R}$, *and* $\Gamma_Q = \Gamma_{Q^{+c}}$ *(here, we take* $\mathcal{U} = \mathcal{U}_Q = \mathcal{U}_{Q^{+c}}$).

*Proof.* By definition $M_Q(u) = \mathbb{E}e^{uY}$ and $M_{Q^{+c}}(u) = \mathbb{E}e^{u(Y+c)}$. Hence,

$$M_{Q^{+c}}(u) = \mathbb{E}e^{u(Y+c)} = e^{uc}\mathbb{E}e^{uY} = e^{uc}M_Q(u).$$

This shows that $\mathcal{U}_Q = \mathcal{U}_{Q^{+c}}$ and that the desired relationship between $M_Q$ and $M_{Q^{+c}}$ hold. Now, from the definition that the CGF is the logarithm of the MGF, it follows that $\psi_{Q^{+c}}(u) = uc + \psi_Q(u)$. Hence, $\ddot{\psi}_{Q^{+c}} = \ddot{\psi}_Q$ and $\dot{\psi}_{Q^{+c}} = \dot{\psi}_Q$, which implies that $\Gamma_Q = \Gamma_{Q^{+c}}$. $\qquad\square$

Thanks to this result, from a bound on the self-concordance function of centered distributions, we can deduce a bound on the self-concordance function of the non-centered ones.

We thus first work on establishing the bound when $Q$ is centered.

Since the theorem statement holds trivially when the variance $\mathrm{Var}(Q)$ of $Q$ is zero, we will also assume with no loss of generality in some of the results below that $Q$ has positive variance.

**Lemma 16.** *Take a distribution* $Q$ *with zero mean and positive variance. Then, there exist* $\eta > 0$ *and* $0 < a \leq b$ *distribution dependent constants such that* $Q([-b, -a]) \geq \eta$.

*Proof.* Since $\dot{\mu}(0) = \mathrm{Var}(Q) > 0$, there exists $a > 0$ and $\alpha > 0$ such that $Q((-\infty, -a])) > \alpha$. As $\lim_{x \to 0} Q((-\infty, -x])) = 0$, we can find some $b > 0$ s.t.

$$Q\left((-\infty, -b]\right) \leq \frac{\alpha}{2}.$$

This implies:

$$Q\left([-b, -a]\right) \geq \frac{\alpha}{2}.$$

The lemma thus holds with $a$ and $b$ described above, and $\eta = \frac{\alpha}{2}$. $\qquad\square$

**Lemma 17.** *Take a distribution $Q$ with zero mean and positive variance. With $\eta, a, b$ as in Lemma 16, for all $u \in \mathcal{U}_Q^\circ$, it holds that*

$$\dot{\mu}(u) \geq a^2 \eta \frac{e^{-ub}}{M_Q(u)}.$$

*Proof.* For $a, b$ described in Lemma 16, we have:

$$
\begin{aligned}
\int_{\mathbb{R}} (y - \mu(u))^2 Q_u(dy) &\geq \int_{-b}^{-a} (y - \mu(u))^2 Q_u(dy), \\
&\geq \int_{-b}^{-a} (y - \mu(0))^2 \frac{\exp(uy)}{M_Q(u)} Q(dy), \\
&\geq a^2 \frac{\exp(-ub)}{M_Q(u)} \int_{-b}^{-a} Q(dy), \\
&\geq \frac{a^2 e^{-ub}}{M_Q(u)} \eta.
\end{aligned}
$$

The first inequality holds as $(y - \mu(u))^2$ is non-negative; the second as $\mu(u)$ increases with $u$; the third as $-b \leq -a < \mu(0)$, $\mu(0) = 0$ and $u \geq 0$; and the last one by Lemma 16. $\qquad\square$

For the upper bound, we present lemmas that bound the (upper and lower) tails of $Q_u$ and the mean $\mu(u)$.

**Lemma 18.** *Take $Q \in \mathcal{E}_{right}(c_1, C_1)$ a centered distribution. Then, for all $0 \leq u < c_1$ and $t \geq 0$, we have*

$$Q_u\left((t, +\infty)\right) \leq \frac{C_1 e}{M_Q(u)} e^{-(c_1 - u)t} \left(1 + \frac{u}{c_1 - u}\right).$$

*Proof.* The inequality is trivially satisfied when $u = 0$. Indeed, in this case $Q = Q_0$, $M_Q(0) = 1$, which together with $Q \in \mathcal{E}_{\mathrm{right}}(c_1, C_1)$ implies the inequality.

Let us now assume that $0 < u < c_1$. Let $v > 0$ be a constant to be chosen later. Then we have that

$$
\begin{aligned}
M_Q(u) Q_u((t, \infty)) &= \int_t^\infty e^{uy} Q(dy) \\
&= \sum_{k=0}^\infty \int_{t+kv}^{t+(k+1)v} e^{uy} Q(dy) \\
&\leq \sum_{k=0}^\infty e^{u(t+kv+v)} \int_{t+kv}^{+\infty} Q(dy) \\
&\leq \sum_{k=0}^\infty e^{u(t+kv+v)} C_1 e^{-c_1(t+kv)} \\
&= C_1 e^{-(c_1-u)t} e^{uv} \sum_{k=0}^\infty e^{-(c_1-u)vk}.
\end{aligned}
$$

We choose $v = 1/u > 0$. Then $M_Q(u)Q_u((t,\infty))$ can be upper bounded by

$$M_Q(u)Q_u((t,\infty)) \leq C_1 e \cdot e^{-(c_1-u)t} \sum_{k=0}^{\infty} e^{-\frac{c_1-u}{u}k}$$

$$= C_1 e \cdot e^{-(c_1-u)t} \frac{1}{1 - e^{-\frac{c_1-u}{u}}}$$

$$\leq C_1 e \cdot e^{-(c_1-u)t} \left(1 + \frac{u}{c_1 - u}\right),$$

where in the last inequality, we used the fact that for all $x > 0$, $\frac{e^x}{e^x-1} \leq 1 + \frac{1}{x}$.  $\square$

**Remark 1.** *For $Q$ a centered exponential distribution $\mathrm{Exp}(c)$ with rate parameter $c$, the moment generating function is $M_Q(u) = e^{-\frac{u}{c}}\frac{c}{c-u}$. On the other hand, $Q_u(t,\infty) = e^{-\frac{u}{c}}e^{-(c-u)t}$. So Lemma 18 is order tight in its dependency on $c - u$.*

**Lemma 19.** *Take $Q \in \mathcal{E}_{left}(c_2, C_2)$ a centered distribution. Then, for all $u, t \geq 0$, we have*

$$Q_u((-\infty, -t)) \leq \frac{1}{M_Q(u)} C_2 e^{-(u+c_2)t}.$$

*Proof.* We again separate the $u = 0$ case. When $u = 0$, $Q = Q_0$, $M_Q(0) = 1$ and the inequality is equivalent to $Q \in \mathcal{E}_{\mathrm{left}}(c_2, C_2)$.

Consider now $u > 0$. Then,

$$M_Q(u)Q_u((-\infty, -t)) = \int_{-\infty}^{-t} e^{uy}Q(dy),$$

$$\leq e^{-ut}\int_{-\infty}^{-t} Q(dy),$$

$$\leq C_2 e^{-ut-c_2t}.$$

$\square$

**Lemma 20.** *Take $Q \in \mathcal{E}_{right}(c_1, C_1)$ with zero mean and positive variance. Define $c_0 = c_1 \cdot C_1 \cdot e$. Then, for all $0 \leq u < c_1$, it holds that*

$$0 = \mu(0) \leq \mu(u) \leq c_0 \left(\frac{1}{c_1 - u}\right)^2.$$

*Proof.* For $u = 0$, $\mu(0) = 0$ which satisfies the inequality. We now consider $0 < u < c_1$. We have

$$\mu(u) = \int_0^{+\infty} Q_u((y, +\infty))\, dy - \int_{-\infty}^0 Q_u((-\infty, -y))\, dy$$

$$\leq \int_0^{+\infty} Q_u((y, +\infty))\, dy.$$

By Lemma 18, this implies:

$$\mu(u) \leq \frac{C_1 e}{M_Q(u)} \int_0^{\infty} e^{-(c_1-u)t}\left(\frac{c_1}{c_1 - u}\right) dt$$

$$= \frac{C_1 e}{M_Q(u)}\left(\frac{c_1}{c_1 - u}\right)\frac{1}{c_1 - u}$$

$$= \frac{c_1 C_1 e}{M_Q(u)}\left(\frac{1}{c_1 - u}\right)^2.$$

By Jensen's inequality, $M_Q(u) = \mathbb{E}_Q[e^{uY}] \geq e^{u\mathbb{E}_Q[Y]} = 1$, finishing the proof.  $\square$

*Proof of Proposition 13.* As noted beforehand, since the statement holds trivially when the variance of $Q$ is zero, we assume it is positive. By our discussion beforehand, we also assume first that $Q$ is centered, so $\mu(0) = 0$.

Let $Y \sim Q_u$. Take $B > 0$ a constant to be optimized later. We have

$$\int |y - \mu(u)|^3 \, Q_u(dy) = \int_0^{+\infty} 3t^2 \, \mathbb{P}\left(|Y - \mu(u)| \geq t\right) dt,$$

$$= \underbrace{\int_0^B 3t^2 \, \mathbb{P}\left(|Y - \mu(u)| \geq t\right) dt}_{(i)} + \underbrace{\int_B^{+\infty} 3t^2 \, \mathbb{P}\left(|Y - \mu(u)| \geq t\right) dt}_{(ii)}.$$

The following bound holds:

$$(i) \leq 3B \int_0^B t \, \mathbb{P}\left(|Y - \mu(u)| \geq t\right) dt,$$

$$\leq \frac{3B}{2} \dot{\mu}(u). \tag{9}$$

We also have:

$$(ii) \leq \underbrace{\int_B^{+\infty} 3t^2 \, \mathbb{P}\left(Y - \mu(u) \geq t\right) dt}_{(ii,a)} + \underbrace{\int_B^{+\infty} 3t^2 \, \mathbb{P}\left(Y \leq -(t - \mu(u))\right) dt}_{(ii,b)}.$$

Set $B = 2c_0 \left(\frac{1}{c_1 - u}\right)^2 + \frac{ub}{c_1 - u} + \frac{ub}{u + c_2}$ and $B' = 2c_0 \left(\frac{1}{c_1 - u}\right)^2$. Then we can upper bound $(ii, a)$ using Lemma 18.

$$(ii, a) \leq \frac{c_0}{M_Q(u)} \left(\frac{1}{c_1 - u}\right) \int_B^\infty 3t^2 e^{-(c_1 - u)(t + \mu(u))} dt$$

$$\leq \frac{c_0}{M_Q(u)} \left(\frac{1}{c_1 - u}\right) \int_B^\infty 3t^2 e^{-(c_1 - u)t} dt$$

$$\leq \frac{c_0 e^{-ub}}{M_Q(u)} \left(\frac{1}{c_1 - u}\right) \int_B^\infty 3t^2 e^{-(c_1 - u)(t - \frac{ub}{c_1 - u})} dt$$

$$= \frac{c_0 e^{-ub}}{M_Q(u)} \left(\frac{1}{c_1 - u}\right) \int_{B - \frac{ub}{c_1 - u}}^\infty 3\left(t + \frac{ub}{c_1 - u}\right)^2 e^{-(c_1 - u)t} dt$$

$$\leq \frac{6c_0 e^{-ub}}{M_Q(u)} \left(\frac{1}{c_1 - u}\right) \int_{B'}^\infty \left[t^2 + \left(\frac{ub}{c_1 - u}\right)^2\right] e^{-(c_1 - u)t} dt$$

$$= \frac{6c_0 e^{-ub}}{M_Q(u)} \left(\underbrace{\left(\frac{1}{c_1 - u}\right) \int_{B'}^\infty t^2 e^{-(c_1 - u)t} dt}_{(iii,a)} + \underbrace{\left(\frac{1}{c_1 - u}\right) \int_{B'}^\infty \left(\frac{ub}{c_1 - u}\right)^2 e^{-(c_1 - u)t} dt}_{(iii,b)}\right)$$

where in the third inequality we used the fact that $(a + b)^2 \leq 2a^2 + 2b^2$ for all $a, b \in \mathbb{R}$ and that $B' < B - \frac{ub}{c_1 - u}$. We now bound $(iii, a)$. We have that $B'(c_1 - u) = 2c_0 \left( \frac{1}{c_1 - u} \right)$ and

$$
\begin{aligned}
(iii, a) &= \frac{1}{(c_1 - u)^4} e^{-B'(c_1 - u)} \left( [B'(c_1 - u) + 1]^2 + 1 \right) \leq \frac{2}{(c_1 - u)^4} e^{-B'(c_1 - u)} (B'(c_1 - u) + 1)^2 \\
&\leq \frac{4}{(c_1 - u)^4} e^{-B'(c_1 - u)} \left( [B'(c_1 - u)]^2 + 1 \right) \\
&= \frac{4}{(c_1 - u)^4} e^{-2c_0 \left( \frac{1}{c_1 - u} \right)} \left( 4c_0^2 \left( \frac{1}{c_1 - u} \right)^2 + 1 \right) \\
&= \frac{16c_0^2}{(c_1 - u)^6} e^{-2c_0 \left( \frac{1}{c_1 - u} \right)} + \frac{4}{(c_1 - u)^4} e^{-2c_0 \left( \frac{1}{c_1 - u} \right)} \\
&\leq 16 \cdot c_0^2 \cdot \frac{1}{c_0^6} \left( c_0 \left( \frac{1}{c_1 - u} \right) \right)^6 \cdot e^{-2c_0 \left( \frac{1}{c_1 - u} \right)} + 4 \cdot \frac{1}{c_0^4} \left( c_0 \left( \frac{1}{c_1 - u} \right) \right)^4 \cdot e^{-2c_0 \left( \frac{1}{c_1 - u} \right)} \\
&\leq \frac{32}{c_0^4} + \frac{2}{c_0^4} \qquad\qquad (x^6 e^{-2x} \leq 2 \text{ and } x^4 e^{-2x} \leq 0.5 \text{ for all } x \geq 0.)
\end{aligned}
$$

Similarly, for $(iii, b)$, we have that

$$
\begin{aligned}
(iii, b) &\leq \frac{2}{c_1 - u} \int_{B'}^{\infty} \left( \frac{u^2 b^2}{(c_1 - u)^2} \right) e^{-(c_1 - u)t} dt \\
&= \frac{2u^2 b^2}{(c_1 - u)^3} \frac{e^{-B'(c_1 - u)}}{c_1 - u} \\
&\leq \frac{2u^2 b^2}{(c_1 - u)^4} e^{-2c_0 \left( \frac{1}{c_1 - u} \right)} \\
&\leq \frac{2u^2 b^2}{c_0^4} \left( \frac{c_0}{c_1 - u} \right)^4 e^{-2c_0 \left( \frac{1}{c_1 - u} \right)} \\
&\leq \frac{u^2 b^2}{c_0^4} \qquad\qquad (x^4 e^{-2x} \leq 0.5 \text{ for all } x \geq 0.)
\end{aligned}
$$

Putting the result together, $(ii, a)$ can be upper bounded as

$$
\begin{aligned}
(ii, a) &\leq \frac{6c_0 e^{-ub}}{M_Q(u)} \left( \frac{32}{c_0^4} + \frac{2}{c_0^4} + \frac{u^2 b^2}{c_0^4} \right) \\
&\leq \frac{e^{-ub}}{M_Q(u)} \left( \frac{204 + 6u^2 b^2}{c_0^3} \right).
\end{aligned}
$$

By Lemma 20, $B \geq \mu(u) + \frac{ub}{c_2 + u}$. Hence By Lemma 19 we have:

$$
\begin{aligned}
(ii, b) &\leq \frac{C_2}{M_Q(u)} \int_B^{+\infty} 3t^2 C_2 e^{-(u+c_2)(t-\mu(u))} dt, \\
&\leq \frac{C_2 e^{-ub}}{M_Q(u)} \int_B^{+\infty} 3t^2 C_2 e^{-(u+c_2)(t-\mu(u) - \frac{ub}{u+c_2})} dt, \\
&\leq \frac{9C_2 e^{-ub}}{M_Q(u)} \int_{B - \mu(u) - \frac{ub}{u+c_2}}^{+\infty} \left( t^2 + \mu(u)^2 + \left( \frac{ub}{u + c_2} \right)^2 \right) e^{-(u+c_2)t} dt.
\end{aligned}
$$

We now focus on

$$
\underbrace{\int_{B - \mu(u) - \frac{ub}{u+c_2}}^{+\infty} t^2 e^{-(u+c_2)t} dt}_{(iv, a)} + \underbrace{\int_{B - \mu(u) - \frac{ub}{u+c_2}}^{+\infty} \mu(u)^2 e^{-(u+c_2)t} dt}_{(iv, b)} + \underbrace{\int_{B - \mu(u) - \frac{ub}{u+c_2}}^{\infty} \left( \frac{ub}{u + c_2} \right)^2 e^{-(u+c_2)t} dt}_{(iv, c)}.
$$

By definition of $B$, we have that $B - \mu(u) - \frac{ub}{u+c_2} \geq c_0 \left(\frac{1}{c_1-u}\right)^2 =: B''$. We then have that

$$(iv, a) \leq \int_{B''}^{+\infty} t^2 e^{-(u+c_2)t} dt$$

$$\leq \frac{e^{-B''(u+c_2)}}{(u+c_2)^3}\left((B''(u+c_2)+1)^2+1\right)$$

$$\leq \frac{2e^{-B''(u+c_2)}}{(u+c_2)^3}(B''(u+c_2)+1)^2$$

$$\leq \frac{4e^{-B''(u+c_2)}}{(u+c_2)^3}([B''(u+c_2)]^2+1)$$

$$\leq \frac{4}{(u+c_2)^3}\left(e^{-B''(u+c_2)}[B''(u+c_2)]^2+e^{-B''(u+c_2)}\right)$$

$$\leq \frac{8}{(u+c_2)^3}$$

For $(iv, b)$, note that $B'' \geq \mu(u)$ by Lemma 20 and we have that

$$(iv, b) \leq \mu(u)^2 \int_{B''}^{\infty} e^{-(u+c_2)t} dt$$

$$\leq \frac{B''^2}{(u+c_2)} e^{-B''(u+c_2)}$$

$$\leq \frac{1}{(u+c_2)^3}(B''(u+c_2))^2 e^{-B''(u+c_2)}$$

$$\leq \frac{1}{(u+c_2)^3}.$$

For $(iv, c)$,

$$(iv, c) \leq \frac{u^2b^2}{(u+c_2)^2} \int_{B''}^{\infty} e^{-(u+c_2)t} dt$$

$$\leq \frac{u^2b^2}{(u+c_2)^2} \frac{1}{u+c_2} e^{-B''(u+c_2)}$$

$$\leq \frac{u^2b^2}{(u+c_2)^3}.$$

Putting bounds on $(iv, a)$, $(iv, b)$ and $(iv, c)$ together, we have that

$$(ii, b) \leq \frac{9C_2 e^{-ub}}{M_Q(u)} \frac{9+u^2b^2}{(u+c_2)^3} \leq \frac{e^{-ub}}{M_Q(u)} \frac{81C_2+9C_2u^2b^2}{(u+c_2)^3}$$

Combining the bounds on $(ii, a)$ and $(ii, b)$ with Lemma 17 we get:

$$\frac{(ii)}{\dot{\mu}(u)} \leq \frac{1}{a^2\eta}\left(\frac{204+6u^2b^2}{c_0^3} + \frac{81C_2+9C_2u^2b^2}{(u+c_2)^3}\right).$$

Chaining the result with the bound on $(i)$ together as well as the choice that $B = 2c_0\left(\frac{1}{c_1-u}\right)^2 + \frac{ub}{c_1-u} + \frac{ub}{u+c_2}$, we obtain

$$\frac{\ddot{\mu}(u)}{\dot{\mu}(u)} \leq \frac{3}{2}\left[2c_0\left(\frac{1}{c_1-u}\right)^2 + \frac{ub}{c_1-u}\right] + \frac{3}{2}b + \frac{1}{a^2\eta}\left(\frac{204+6u^2b^2}{c_0^3} + \frac{81C_2+9C_2u^2b^2}{(u+c_2)^3}\right)$$

$$\leq \frac{3}{2}\left[2c_0\left(\frac{1}{c_1-u}\right)^2 + \frac{ub}{c_1-u} + \frac{ub}{u+c_2}\right] + \underbrace{\frac{1}{a^2\eta}\left(\frac{204+6c_1^2b^2}{(c_1\cdot C_1\cdot e)^3} + \frac{81C_2+9C_2c_1^2b^2}{c_2^3}\right)}_{=G_Q(C_1,C_2,c_1,c_2)}.$$

Let us now study the case $\mu_Q(0) \neq 0$. Let $a, b, \eta > 0$ be such that $Q([-b + \mu(0), -a + \mu(0)]) > \eta$ and $-a < 0$. With $Q^{-\mu(0)}$ the centered version of $Q$, this gives $Q^{-\mu(0)}([-b, -a]) > \eta$. We have just shown that for all $u \in [0; c_1)$,:

$$\Gamma_{Q^{-\mu(0)}}(u) \leq \frac{3}{2}\left[2c_0\left(\frac{1}{c_1 - u}\right)^2 + \frac{ub}{c_1 - u} + \frac{ub}{u + c_2}\right] + G_Q(C_1, C_2, c_1, c_2).$$

By Lemma 15, we have $\Gamma_{Q^{-\mu(0)}} = \Gamma_Q$, hence:

$$\Gamma_Q(u) \leq \frac{3}{2}\left[2c_0\left(\frac{1}{c_1 - u}\right)^2 + \frac{ub}{c_1 - u} + \frac{ub}{u + c_2}\right] + G_Q(C_1, C_2, c_1, c_2).$$

$\square$

## C.2  Proof of Corollary 14

**Lemma 21.** *Let $Y \sim Q$, and $Q^-$ let be the distribution of $-Y$. Then $\mathcal{U}_Q = -\mathcal{U}_{Q^-}$ and for any $u \in \mathcal{U}_Q^\circ$, we have*

$$\Gamma_Q(u) = \Gamma_{Q^-}(-u).$$

*Proof.* Recall that if $Q$ has zero variance, $\Gamma_Q \equiv 0$ and hence the statement is trivial. Otherwise, for $u \in \mathcal{U}_Q^\circ$, $\Gamma_Q(u) = |\dddot{\psi}_Q(u)|/\ddot{\psi}_Q(u)$. Now, for $v \in \mathbb{R}$,

$$M_{Q^-}(v) = \mathbb{E}[e^{(-Y)v}] = \mathbb{E}[e^{(-v)Y}] = M_Q(-v).$$

Hence, $\mathcal{U}_Q = -\mathcal{U}_{Q^-}$ and for any $v \in \mathcal{U}_Q$, $\psi_Q(v) = \psi_{Q^-}(-v)$. Taking derivatives of both sides,

$$\dot{\psi}_Q(v) = -\dot{\psi}_{Q^-}(-v),$$
$$\ddot{\psi}_Q(v) = \ddot{\psi}_{Q^-}(-v),$$
$$\dddot{\psi}_Q(v) = -\dddot{\psi}_{Q^-}(-v),$$

which immediately implies the statement, noting that the variance of $Q$ is positive if and only if the variance of $Q^-$ is positive. $\square$

*Proof of Corollary 14.* Assume that $Q \in \mathcal{E}_{\text{right}}(c_1, C_1) \cap \mathcal{E}_{\text{left}}(c_2, C_2)$. Since the statement holds trivially when $\text{Var}(Q) = 0$, assume $\text{Var}(Q) > 0$. Then, $Q^- \in \mathcal{E}_{\text{right}}(c_2, C_2) \cap \mathcal{E}_{\text{right}}(c_1, C_1)$. We then get the stated result by applying Proposition 13, combined with Lemma 21. To be more specific, for all $u \in (-c_2, 0]$, from these two results it follows that

$$\Gamma_Q(u) = \Gamma_{Q^-}(-u)$$
$$\leq \frac{3}{2}\left[2c_0\left(\frac{1}{c_2 - |u|}\right)^2 + \frac{|u|b}{c_2 - |u|} + \frac{|u|}{c_1 + |u|}b\right] + \frac{1}{a^2\eta}\left(\frac{204 + 6u^2b^2}{c_0^3} + \frac{81C_1 + 9C_1u^2b^2}{(u + c_1)^3}\right)$$
$$\leq \frac{3}{2}\left[2c_0\left(\frac{1}{c_2 - |u|}\right)^2 + \frac{|u|b}{c_2 - |u|} + b\right] + \frac{1}{a^2\eta}\left(\frac{204 + 6c_2^2b^2}{c_0^3} + \frac{81C_1 + 9C_1c_2^2b^2}{c_1^3}\right),$$

where $c_0 = C_2 \cdot c_2 \cdot e$.

$\square$

## C.3  Proof of Corollary 8

Let $\mathcal{Q}$ be a regular NEF with base distribution $Q$. By definition, this means that $\mathcal{U}_Q$ is an open interval. Take any $u \in \mathcal{U}_Q^\circ$. There exist some $\epsilon > 0$ s.t. $M_Q(u - \epsilon), M_Q(u + \epsilon) < \infty$. We also have:

$$M_Q(u + \epsilon) = \int \exp(\epsilon u)\exp(uy)Q(dy)$$
$$= \int \exp(\epsilon u)M_Q(u)Q_u(dy)$$
$$= M_Q(u)M_{Q_u}(\epsilon).$$

Similarly, $M_{Q_u}(-\epsilon) = \frac{M_Q(u-\epsilon)}{M_Q(u)}$. By Proposition 6, this implies $Q_u \in \mathcal{E}_{\text{left}}\left(\frac{M_Q(u-\epsilon)}{M_Q(u)}, \epsilon\right) \cap \mathcal{E}_{\text{right}}\left(\frac{M_Q(u+\epsilon)}{M_Q(u)}, \epsilon\right)$.

## D   Self concordance for subgaussian base distributions

For the convenience of the reader we restate the theorems to be proven.

**Theorem 9.** *Let $Q$ be subgaussian. Then, the NEF $\mathcal{Q} = (Q_u)_{u \in \mathbb{R}}$ is self-concordant and $\Gamma_Q(u) = O(|u|)$, $u \in \mathbb{R}$.*

**Theorem 10.** *There exists a distribution $Q$ that is subgaussian such that $\limsup_{u \to \infty} \Gamma_Q(u)/u > 0$.*

We start by introducing some notations reminiscent of the ones used for subexponential distributions. Recall that a centered distribution $Q$ is subgaussian if and only if for some $c, C > 0$, it holds that

$$\mathbb{P}(|X| \geq t) \leq C \exp(-\frac{ct^2}{2}) \qquad \text{for all } t \geq 0 \tag{10}$$

where $X \sim Q$. Let

$$\mathcal{G}(c, C) = \{Q \, : \, Y = X - \mathbb{E}X \text{ satisfies Eq. (10), where } X \sim Q\}.$$

This definition is very close to that of $\mathcal{E}_{\text{right}}$. In words, $\mathcal{G}$ is the class of distributions over the reals whose left and right-tail display a subgaussian decay governed with the rate parameter $c > 0$ and scaling constant $C > 0$.

For the interested reader, we also report here without proof some classical results on the quantitative relation between the $MGF$ and the tail bounds of subgaussian distributions. Details can be found in the textbook [RH23].

**Proposition 22** ([RH23]). *If for all $u \in \mathbb{R}, M_Q(u) \leq e^{\sigma^2 u^2/2}$, then $\mathbb{P}(|X| \geq t) \leq 2 \exp(-t^2/(2\sigma^2))$, and if $\mathbb{P}(|X| \geq t) \leq 2 \exp(-t^2/(2\sigma^2))$, then for all $u \in \mathbb{R}$, $M_Q(u) \leq e^{4\sigma^2 u^2}$.*

### D.1   Proof of Theorem 9

Lemma 12 still holds for $Q$ a subgaussian distribution, hence the theorem holds trivially for $\dot{\mu}(0) = 0$. Lemma 15 can also be applied for $Q$ a subgaussian distribution. Thus, as in the proof for subexponential distributions, the first step is to show the result when $\mu(0) = 0$ and $\dot{\mu}(0) > 0$.

Note that the lower bound on $\dot{\mu}(u)$ of Lemma 17 still holds when $\dot{\mu}(0) > 0$. We turn to upper bounding $\int |y - \mu(u)|^3 Q_u(dy)$. The following Lemmas are first steps in that direction.

**Lemma 23.** *Take $Q \in \mathcal{G}(c, C)$ a centered distribution. For all $u \geq 0$ and $t \geq \left(\frac{4}{c} + 1\right) u + \frac{4}{c}$, we have:*

$$Q_u((t, +\infty)) \leq \frac{\mathfrak{C}_1}{M_Q(u)} e^{-\frac{ct^2}{4}},$$

*where $\mathfrak{C}_1 = C\left(1 + \sqrt{\frac{\pi}{c}}\right)$.*

*Proof.* We have

$$M_Q(u)Q_u\left((t,+\infty)\right) = \int_t^{+\infty} e^{uy}Q(dy),$$

$$= \sum_{k=0}^{+\infty} \int_{t+k}^{t+k+1} e^{uy}Q(dy),$$

$$\leq \sum_{k=0}^{+\infty} e^{u(t+k+1)} \int_{t+k}^{t+k+1} Q(dy),$$

$$\leq \sum_{k=0}^{+\infty} e^{u(t+k+1)}Q\left([t+k;+\infty)\right),$$

$$\leq C\sum_{k=0}^{+\infty} e^{-\left(\frac{c(t+k)^2}{2}-u(t+k+1)\right)}. \tag{11}$$

The first inequality holds as $u \geq 0$, the second by subgaussianity assumption. As $t \geq \left(\frac{4}{c}+1\right)u + \frac{4}{c}$, we have $u \leq \frac{c}{4}t$ and:

$$t^2 \geq \left(\frac{4}{c}+1\right)ut = \frac{4t}{c}u + \underbrace{t}_{\geq \frac{4}{c}}u \geq \frac{4}{c}(t+1)u. \tag{12}$$

This implies $u(t+1) \leq \frac{c}{4}t^2$ on top of $u \leq \frac{c}{4}t$. Then

$$u(t+1+k) \leq \frac{c}{4}\left(t^2+kt\right) \leq \frac{c}{4}\left(t+k\right)^2.$$

Reinjecting in Eq. (11), we get:

$$M_Q(u)Q_u\left((t,+\infty)\right) \leq C\sum_{k=0}^{+\infty} e^{-\frac{c(t+k)^2}{4}},$$

$$\leq Ce^{-\frac{ct^2}{4}}\sum_{k=0}^{+\infty} e^{-\frac{ck^2}{4}}.$$

$$\leq Ce^{-\frac{ct^2}{4}}\left(1+\int_0^\infty e^{-\frac{cx^2}{4}}dx\right)$$

$$\leq C\left(1+\sqrt{\frac{\pi}{c}}\right)e^{-\frac{ct^2}{4}}.$$

$\square$

**Lemma 24.** *Take $Q \in \mathcal{G}(c,C)$ a centered distribution. For all $u, t \geq 0$, we have:*

$$Q_u\left((-\infty,-t)\right) \leq \frac{C}{M_Q(u)}e^{-ut-\frac{ct^2}{2}}.$$

*Proof.* We have that

$$M_Q(u)Q_u\left((-\infty,-t)\right) = \int_{-\infty}^{-t} e^{uy}Q(dy),$$

$$\leq e^{-ut}\int_{-\infty}^{-t} Q(dy),$$

$$\leq Ce^{-ut-\frac{ct^2}{2}}, \tag{13}$$

where the last inequality holds by subgaussianity of $Q$. $\square$

**Lemma 25.** *Take $Q \in \mathcal{G}(c, C)$ a centered distribution. For all $u \geq 0$ the following holds:*

$$0 = \mu(0) \leq \mu_Q(u) \leq \left(\frac{4}{c} + 1\right) u + \frac{4}{c} + \mathfrak{C}_3 e^{-\frac{4u^2}{c}},$$

*where $\mathfrak{C}_3 = \frac{\sqrt{\pi}}{\sqrt{c}} \mathfrak{C}_1$.*

*Proof.* We have:

$$\mu(u) = \int_0^{+\infty} Q_u\left((y, +\infty)\right) dy - \int_{-\infty}^0 Q_u\left((-\infty, -y)\right) dy$$

$$\leq \int_0^{+\infty} Q_u\left((y, +\infty)\right) dy$$

$$\leq \left(\frac{4}{c} + 1\right) u + \frac{4}{c} + \int_{\left(\frac{4}{c}+1\right)u+\frac{4}{c}}^{+\infty} Q_u\left((y, +\infty)\right) dy.$$

By Lemma 23, this implies:

$$\mu(u) \leq \left(\frac{4}{c} + 1\right) u + \frac{4}{c} + \frac{\mathfrak{C}_1}{M_Q(u)} \int_{\left(\frac{4}{c}+1\right)u+\frac{4}{c}}^{+\infty} e^{-\frac{cy^2}{4}} dy$$

$$\leq \left(\frac{4}{c} + 1\right) u + \frac{4}{c} + \frac{\mathfrak{C}_1 e^{-c\frac{\left(\left(\frac{4}{c}+1\right)u+\frac{4}{c}\right)^2}{4}}}{M_Q(u)} \int_0^{+\infty} e^{-\frac{cy^2}{4}} dy$$

$$\leq \left(\frac{4}{c} + 1\right) u + \frac{4}{c} + \frac{\mathfrak{C}_1}{M_Q(u)} e^{-\frac{4u^2}{c}} \int_0^{+\infty} e^{-\frac{cy^2}{4}} dy.$$

By Jensen's inequality, $M_Q(u) = \mathbb{E}_Q[e^{uY}] \geq e^{u\mathbb{E}_Q[Y]} = 1$. Noting that $\int_0^{+\infty} e^{-\frac{cy^2}{4}} dy \leq \frac{\sqrt{\pi}}{\sqrt{c}}$ terminates the proof. $\square$

*Proof of Theorem 9.* As previously noted, we first show the result when $\mu(0) = 0$ and $\dot{\mu}(0) > 0$. We start with $u > 0$, then by Lemma 21 will extend the bound to $u < 0$.

Take $u > 0$ and $B > 0$ a constant to be optimized later. We have:

$$\int |Y - \mu_Q(u)|^3 Q_u(dy) = \int_0^{+\infty} 3t^2 \, \mathbb{P}\left(|Y - \mu_Q(u)| \geq t\right) dt,$$

$$\leq \underbrace{\int_0^B 3t^2 \, \mathbb{P}\left(|Y - \mu_Q(u)| \geq t\right) dt}_{(i)} + \underbrace{\int_B^{+\infty} 3t^2 \, \mathbb{P}\left(|Y - \mu_Q(u)| \geq t\right) dt}_{(ii)}.$$

The first equality is a classical result on the relationship between moments and tails of r.v., see for instance Exercise 1.2.3 of [Ver18]. The following bound holds:

$$(i) \leq 3B \int_0^B t \, \mathbb{P}\left(|Y - \mu_Q(u)| \geq t\right) dt,$$

$$\leq \frac{3B}{2} \dot{\mu}_Q(u). \tag{14}$$

We also have:

$$(ii) \leq \underbrace{\int_B^{+\infty} 3t^2 \, \mathbb{P}\left(Y - \mu_Q(u) \geq t\right) dt}_{(ii,a)} + \underbrace{\int_B^{+\infty} 3t^2 \, \mathbb{P}\left(Y \leq -(t - \mu_Q(u))\right) dt}_{(ii,b)}.$$

Set $B = \mu_Q(u) + \left(\frac{4}{c} + 1\right) u + \frac{4}{c} + b$, where $b$ was defined in Lemma 17. As $B \geq \left(\frac{4}{c} + 1\right) u + \frac{4}{c}$, by Lemma 23, we have:

$$(ii, a) \leq \int_B^{+\infty} 3t^2 \, Q_u \left((t, +\infty)\right) dt,$$

$$\leq \frac{\mathfrak{C}_1}{M_Q(u)} \int_B^{+\infty} 3t^2 e^{-\frac{ct^2}{4}} dt,$$

$$\leq \frac{3\mathfrak{C}_1}{M_Q(u)} \int_0^{+\infty} (B+t)^2 e^{-\frac{c(B+t)^2}{4}} dt,$$

$$\leq \frac{6\mathfrak{C}_1 e^{-\frac{cB^2}{4}}}{M_Q(u)} \int_0^{+\infty} (B^2 + t^2) e^{-\frac{ct^2}{4}} dt,$$

$$= \frac{6\mathfrak{C}_1 e^{-\frac{cB^2}{4}}}{M_Q(u)} (B^2 + \frac{2}{c}) \int_0^{+\infty} e^{-\frac{ct^2}{4}} dt,$$

$$= \frac{6\mathfrak{C}_1}{M_Q(u)} \sqrt{\frac{\pi}{c}} (B^2 + \frac{2}{c}) e^{-\frac{cB^2}{4}}.$$

where the first equality holds as for any $a > 0$, using integration by part we have that

$$\int_0^{+\infty} e^{-at^2} dt = 2a \int_0^{+\infty} t^2 e^{-at^2 dt}. \tag{15}$$

Denote $B' = B - \mu_Q(u) = \left(\frac{4}{c} + 1\right) u + \frac{4}{c} + b$. By Lemma 24 we have:

$$(ii, b) \leq \int_B^{+\infty} 3t^2 \frac{\mathfrak{C}_2}{M_Q(u)} e^{-u(t - \mu_Q(u)) - \frac{c(t - \mu_Q(u))^2}{2}} dt,$$

$$= \frac{3\mathfrak{C}_2}{M_Q(u)} \int_{B'}^{+\infty} (t + \mu_Q(u))^2 e^{-ut - \frac{ct^2}{2}} dt,$$

$$\leq \frac{3\mathfrak{C}_2 e^{-\frac{4}{c} u^2 - bu}}{M_Q(u)} \int_{B'}^{+\infty} (t + \mu_Q(u))^2 e^{-\frac{ct^2}{2}} dt,$$

$$\leq \frac{3\mathfrak{C}_2 e^{-\frac{4}{c} u^2 - bu}}{M_Q(u)} e^{-\frac{c(B')^2}{2}} \int_0^{+\infty} (t + B' + \mu_Q(u))^2 e^{-\frac{ct^2}{2}} dt,$$

$$\leq \frac{6\mathfrak{C}_2 e^{-\frac{4}{c} u^2 - bu}}{M_Q(u)} e^{-\frac{c(B')^2}{2}} \int_0^{+\infty} (t^2 + B^2) e^{-\frac{ct^2}{2}} dt,$$

$$\leq \frac{6\mathfrak{C}_2 e^{-\frac{4}{c} u^2 - bu}}{M_Q(u)} e^{-\frac{c(B')^2}{2}} \left(\frac{1}{c} + B^2\right) \int_0^{+\infty} e^{-\frac{ct^2}{2}} dt,$$

$$\leq \frac{6 e^{-\frac{12}{c} u^2 - bu}}{M_Q(u)} \left(\frac{1}{c} + B^2\right) \underbrace{\int_0^{+\infty} \mathfrak{C}_2 e^{-\frac{ct^2}{2}} dt}_{\mathfrak{C}_4}.$$

where in the second line we use change of variable; third line we use the fact that $B' > \frac{4}{c} u + b$, hence $e^{-ut} \leq e^{-B'u} \leq e^{-\frac{4}{c} u^2 - ub}$ for all $t \geq B'$; fourth line change of variable and the fact that $-(a + b)^2 \leq -a^2 - b^2$ for all $a, b \geq 0$; fifth line the fact that $(a + b)^2 \leq 2a^2 + 2b^2$ for all $a, b \in \mathbb{R}$; sixth line Eq. (15); and seventh line $B' > \frac{4}{c} u$.

Combining the bounds on $(ii, a)$ and $(ii, b)$ with Lemma 17 we get:

$$\frac{(ii)}{\dot{\mu}_Q(u)} \leq \frac{6}{a^2 \eta} \left(\mathfrak{C}_4 \left(\frac{1}{c} + B^2\right) e^{-\frac{12}{c} u^2} + \mathfrak{C}_3 \left(\frac{2}{c} + B^2\right) e^{-\frac{cB^2}{4} + ub}\right).$$

As $B \geq \frac{4}{c} u + b$, we have $\frac{cB^2}{4} + ub \geq \frac{4}{c} u^2$. This implies

$$\frac{(ii)}{\dot{\mu}_Q(u)} \leq \frac{6(\mathfrak{C}_3 + \mathfrak{C}_4)}{a^2 \eta} \left(\frac{2}{c} + B^2\right) e^{-\frac{4}{c} u^2}.$$

With Equation 14, we get

$$\Gamma_Q(u) \le \frac{6(\mathfrak{C}_3 + \mathfrak{C}_4)}{a^2\eta}\left(\frac{2}{c} + B^2\right)e^{-\frac{4}{c}u^2} + \frac{3B}{2}.$$

By Lemma 25, $\mu_Q(u) \le \left(\frac{4}{c} + 1\right)u + \frac{4}{c} + \mathfrak{C}_3$, which implies :

$$B \le 2\left(\frac{4}{c} + 1\right)u + \frac{8}{c} + b + \mathfrak{C}_3.$$

For any constants $w_1, w_2 > 0$, we have $w_1 u^2 e^{-w_2 u^2} \le \frac{w_1}{\sqrt{w_2}}u$. This implies that for $u > 0$:

$$\Gamma_Q(u) = O(u).$$

Note that if $Q \in \mathcal{G}(c, C)$, with $Q^-$ the distribution of $-Y$, $Y \sim Q$, we have $Q^- \in \mathcal{G}(c, C)$. By Lemma 21, that implies:

$$\Gamma_Q(-u) = \Gamma_{Q^-}(u) = O(u).$$

Therefore, for $u \in \mathbb{R}$:

$$\Gamma_Q(u) = O(|u|).$$

As, by Lemma 12, $\Gamma_Q$ is constant for $\dot\mu(0) = 0$, it remains now only to show the theorem if $\mu(0) \ne 0$ and $\dot\mu(0) > 0$. We have just shown

$$\Gamma_{Q^{-\mu(0)}}(u) = O(|u|),$$

with $Q^{-\mu(0)}$ the centered version of $Q$. by Lemma 15, we have $\Gamma_Q = \Gamma_{Q^{-\mu(0)}}$. Hence, for any $Q \in \mathcal{G}(c, C)$:

$$\Gamma_Q(u) = O(|u|).$$

$\square$

### D.2  Proof for Theorem 10

We construct a distribution $Q$ s.t. for any $s > 0$, we can find some $u > s$ with :

$$\frac{\int (y - \mu(u))^3 Q_u(dy)}{\int (y - \mu(u))^2 Q_u(dy)} \ge 0.038u.$$

Let $Q$ the base measure be

$$Q(dy) = \sum_{i \ge 1} p_i \delta_{2^i}(dy), \tag{16}$$

with

$$p_i = \begin{cases} C_1\exp(-4^i) & \text{if } i \text{ is even}, \\ \frac{C_1}{4}\exp(-3 \times 4^{i-1}) & \text{if } i \text{ is odd}, \end{cases} \tag{17}$$

where $C_1$ is a normalizing constant. By definition, we have that $Q_u(dy) = \sum_{i \ge 1} q_i \delta_{2^i}(dy)$ where

$$q_k = \begin{cases} \frac{C_1}{M_Q(u)}\exp(u2^k)\exp(-4^k) & \text{if } k \text{ is even} \\ \frac{C_1}{4M_Q(u)}\exp(u2^k)\exp(-3 \times 4^{k-1}) & \text{if } k \text{ is odd}. \end{cases} \tag{18}$$

We are going to inspect $\frac{|\mathbb{E}[(Y-\mu(u))^3]|}{\mathbb{E}[(Y-\mu(u))^2]}$ for $i$ a positive even number large enough and $u = 2^{i+1}$. By definition, we have that $Q_u(dy) = \sum_{i \ge 1} q_i \delta_{2^i}(dy)$ where

$$q_k = \begin{cases} \frac{C_1}{M_Q(u)}\exp(2^{i+k+1})\exp(-4^k) & \text{if } k \text{ is even}, \\ \frac{C_1}{4M_Q(u)}\exp(2^{i+k+1})\exp(-3 \times 4^{k-1}) & \text{if } k \text{ is odd}. \end{cases} \tag{19}$$

First, we remark a few simple equalities and inequalities that will prove useful in the subsequent computations.

Not that $\mathbb{P}\left(U = 2^i\right) = \frac{C_1}{M_Q(u)} e^{4^i}$, hence:

$$\frac{\mathbb{P}\left(U = 2^{i+1}\right)}{\mathbb{P}\left(U = 2^i\right)} = \frac{1}{4} e^{2^{2(i+1)} - 3 \times 4^i - 4^i} = \frac{1}{4}. \tag{20}$$

This implies:

$$\mathbb{P}\left(U = 2^i\right) \leq \frac{4}{5}, \tag{21}$$

and, combining with Equation Eq. (19),

$$\frac{\mathbb{P}\left(U = 2^j\right)}{\mathbb{P}\left(U = 2^i\right)} = \begin{cases} e^{2^{i+j+1} - 4^j - 4^i} & \text{if } j \text{ is even,} \\ \frac{1}{4} e^{2^{i+j+1} - 3 \times 4^{j-1} - 4^i} & \text{if } j \text{ is odd.} \end{cases} \tag{22}$$

From this last equation we get upper bound:

$$\frac{\mathbb{P}\left(U = 2^j\right)}{\mathbb{P}\left(U = 2^i\right)} \leq e^{2^{i+j+1} - 3 \times 4^{j-1} - 4^i}. \tag{23}$$

The two next Lemma combined show that for any even $i \geq 4$, $1.24 \times 2^i \leq \mu(u) \leq 1.26 \times 2^i$.

**Lemma 26.** *Let $i \geq 4$ be an even number and $u = 2^{i+1}$. For random variable $U \sim Q_u(dy)$, it holds that:*

$$\mu(u) \geq 1.24 \times 2^i. \tag{24}$$

*Proof.* Consider $j < i$, $j$ even. By Eq. (23),

$$\frac{\mathbb{P}\left(U = 2^j\right)}{\mathbb{P}\left(U = 2^i\right)} \leq e^{2^{i+j+1} - 3 \times 4^{j-1} - 4^i}.$$

Denote $f_1(j) = 2^{i+j+1} - 3 \times 4^{j-1} - 4^i$. The monotonicity of the right hand side, $e^{f_1(j)}$, is the same as $f_1(j)$. Take derivative of $f_1(j)$, we have that

$$f_1'(j) = 2^j \cdot 2^{i+1} \ln 2 - 3 \times 4^{j-1} \ln 4 = 2^j (2^{i+1} \ln 2 - 3 \times 2^{j-2} \ln 4) = 2^j \ln 4 (2^i - 3 \times 2^{j-2}).$$

Since $i > j$ we have that $f_1'(j) \geq 0$ for $j \in [0, i]$. Hence the right hand side increases with $j$, as $j \leq i - 1$, we have that

$$\frac{\mathbb{P}\left(U = 2^j\right)}{\mathbb{P}\left(U = 2^i\right)} \leq e^{-3 \times 4^{i-2}}. \tag{25}$$

This implies:

$$\frac{\mathbb{P}\left(U < 2^i\right)}{\mathbb{P}\left(U = 2^i\right)} = \frac{\sum_{j<i} \mathbb{P}\left(U = 2^j\right)}{\mathbb{P}\left(U = 2^i\right)} \leq (i-1)e^{-3 \times 4^{i-2}} \leq 0.001. \tag{26}$$

We now prove the lower bound on the mean.

$$\begin{aligned} \mu(u) \geq & 2^{i+1}\mathbb{P}(U = 2^{i+1}) + 2^i\mathbb{P}(U = 2^i) + 2^i\mathbb{P}(U > 2^{i+1}) \\ = & 2^{i+1}\mathbb{P}(U = 2^{i+1}) + 2^i\mathbb{P}(U = 2^i) + 2^i(1 - \mathbb{P}(U < 2^i) - \mathbb{P}(U = 2^i) - \mathbb{P}(U = 2^{i+1})) \\ \geq & 2^i + \left[2^{i+1} - 2^i\right]\mathbb{P}(U = 2^{i+1}) - 2^i\mathbb{P}(U < 2^i) \\ = & 2^i + 2^i\mathbb{P}(U = 2^i)\left(\frac{\mathbb{P}(U = 2^{i+1})}{\mathbb{P}(U = 2^i)} - \frac{\mathbb{P}(U < 2^i)}{\mathbb{P}(U = 2^i)}\right) \\ \geq & 2^i + 2^i\left(\frac{1}{4} - 0.001\right)\mathbb{P}(U = 2^i) \\ \geq & 1.24 \times 2^i, \end{aligned}$$

where the fourth line holds because of Eqs. (20) and (26). □

**Lemma 27.** *Let $i$ be a positive even number and $u = 2^{i+1}$. For random variable $U \sim Q_u(dy)$, it holds that:*

$$\mu(u) \leq 1.26 \cdot 2^i. \tag{27}$$

*Proof.* By Eq. (23), for any $j \geq i + 2$, we have:

$$\frac{\mathbb{P}\left(U = 2^j\right)}{\mathbb{P}\left(U = 2^i\right)} \leq e^{2^{i+j+1} - 3 \times 4^{j-1} - 4^i} = e^{-2^{j-1}\left(3 \times 2^{j-1} - 2 \times 2^{i+1}\right) - 4^i},$$

$$\leq e^{-4^i - 2^{j-1}}. \tag{28}$$

This implies:

$$\sum_{j>i+1} 2^j \frac{\mathbb{P}_u\left(U = 2^j\right)}{\mathbb{P}\left(U = 2^i\right)} \leq e^{-4^i} \sum_{j>i+1} 2^j e^{-2^{j-1}} \leq \left(\sum_{j=4}^{\infty} 2^j e^{-2^{j-1}}\right) e^{-4^i}.$$

We turn to bounding $\sum_{j=4}^{\infty} 2^j e^{-2^{j-1}}$:

$$\sum_{j=4}^{\infty} 2^j e^{-2^{j-1}} = 2\sum_{j=4}^{\infty} 2^{j-1} e^{-2^{j-1}} = 2\sum_{j=3}^{\infty} 2^j e^{-2^j},$$

$$\leq 2\int_2^{\infty} 2^x e^{-2^x} dx,$$

$$= 2\int_2^{\infty} \frac{1}{y \ln 2} y e^{-y} dy,$$

$$= \frac{2}{e^4 \ln 2},$$

where the second inequality holds as $2^x e^{-2^x}$ is decreasing for $x > 0$. Hence, for any $i \geq 2$:

$$\sum_{j>i+1} 2^j \frac{\mathbb{P}_u\left(U = 2^j\right)}{\mathbb{P}\left(U = 2^i\right)} \leq \frac{2}{e^4 \ln 2} e^{-4^i} \leq \frac{1}{100} 2^i. \tag{29}$$

We are now ready to upper bound $\mu_u$:

$$\mu_u \leq 2^{i-1}\mathbb{P}(U < 2^i) + 2^i \mathbb{P}(U = 2^i) + 2^{i+1}\mathbb{P}(U = 2^{i+1}) + \sum_{j>i+1} 2^j \mathbb{P}(U = 2^j)$$

$$\leq 2^i P(U = 2^i)\left(\frac{1}{2}\frac{P(U < 2^i)}{P(U = 2^i)} + 1 + 2\frac{P(U = 2^{i+1})}{P(U = 2^i)} + \frac{1}{2^i}\sum_{j>i+1} 2^j \frac{\mathbb{P}_u\left(U = 2^j\right)}{\mathbb{P}\left(U = 2^i\right)}\right)$$

$$\leq 2^i \frac{4}{5}\left(\frac{1}{2} \times 0.001 + 1 + \frac{1}{2} + \frac{1}{100}\right)$$

$$< 1.26 \cdot 2^i,$$

where we get the third inequality from Eqs. (20), (21), (26) and (29).

$\square$

*Proof for Theorem 10.* Let $U \sim Q_u$ where $u = 2^{i+1}$ for $i \geq 4$ an even number. We first derive an upper bound on the variance $\mathbb{E}[(U - \mu(u))^2] \leq \mathbb{E}[U^2]$.

$$\mathbb{E}[U^2] \leq 2^{2i} \cdot \mathbb{P}(U < 2^i) + 2^{2i} \cdot \mathbb{P}(U = 2^i) + 2^{2i+2} \cdot \mathbb{P}(U = 2^{i+1}) + \sum_{j > i+1} 2^{2j} \mathbb{P}(U = 2^j)$$

$$\leq 2^{2i} P(U = 2^i) \left( \frac{\mathbb{P}(U < 2^i)}{\mathbb{P}(U = 2^i)} + 1 + 4 \frac{\mathbb{P}(U = 2^{i+1})}{\mathbb{P}(U = 2^i)} + \frac{1}{2^{2i}} \sum_{j > i+1} 2^{2j} \frac{\mathbb{P}(U = 2^j)}{\mathbb{P}(U = 2^i)} \right)$$

$$\leq 2^{2i} \frac{4}{5} \left( 0.112 + 1 + 1 + \frac{e^{-4^i}}{2^{2i}} \sum_{j > i+1} 2^{2j} e^{-2^{j-1}} \right), \tag{30}$$

where the last inequality is obtained from Eqs. (20), (21), (26) and (28).

We inspect the infinite series in the above inequality . Note that $2^{2j} e^{-2^{j-1}}$ is decreasing for $j \geq 1$.

$$\sum_{j > i+1} 2^{2j} e^{-2^{j-1}} \leq 4 \sum_{j \geq 2} 4^j e^{-2^j}$$

$$\leq 4 \int_1^\infty 4^x e^{-2^x} dx$$

$$= 4 \int_2^\infty \frac{1}{y \ln 2} y^2 e^{-y} dy$$

$$= \frac{12}{\ln 2 \cdot e^2}.$$

For all $i \geq 1$, it follows that

$$\frac{e^{-4^i}}{2^{2i}} \frac{12}{\ln 2 \cdot e^2} \leq 0.02.$$

Plugging into Equation 30, we have:

$$\mathbb{E}[(U - \mu(u))^2] \leq 1.7 \times 2^{2i}, \tag{31}$$

On the other hand, by Lemmas 26 and 27, we have that

$$U - \mu(u) \geq \begin{cases} \underbrace{0.74 \cdot 2^i}_{A_{\text{large}}} & \text{if } U \geq 2^{i+1} \\ \underbrace{-0.26 \cdot 2^i}_{A_{\text{medium}}} & \text{if } U = 2^i \\ \underbrace{-1.26 \cdot 2^i}_{A_{\text{small}}} & \text{if } U < 2^i. \end{cases}$$

Then we can lower bound the third central moment $\mathbb{E}[|U - \mu(u)|^3]$.

$$\mathbb{E}[(U - \mu(u))^3] \geq \mathbb{P}(U \geq 2^{i+1}) A_{\text{large}}^3 + \mathbb{P}(U = 2^i) A_{\text{medium}}^3 + \mathbb{P}(U < 2^i) A_{\text{small}}^3$$

$$= [1 - \mathbb{P}(U = 2^i) - \mathbb{P}(U < 2^i)] \cdot A_{\text{large}}^3 + \mathbb{P}(U = 2^i) A_{\text{medium}}^3 + \mathbb{P}(U < 2^i) A_{\text{small}}^3$$

$$= (0.74 \cdot 2^i)^3 - \mathbb{P}(U = 2^i)[(0.74 \cdot 2^i)^3 + (0.26 \cdot 2^i)^3] - \mathbb{P}(U < 2^i)((0.74 \cdot 2^i)^3 + (1.26 \cdot 2^i)^3)$$

$$\geq (2^i)^3 \left[ 0.74^3 - 0.8(0.74^3 + 0.26^3) - 0.8 \times 0.001(1.26^3 + 0.74^3) \right]$$

$$\geq 0.065(2^i)^3.$$

where the third inequality holds by Eqs. (21) and (26). Combining this last bound with Eq. (31), we obtain:

$$\frac{\mathbb{E}[(U - \mu(u))^3]}{\mathbb{E}[(U - \mu(u))^2]} \geq \frac{0.065}{1.7} 2^i = 0.038u.$$

$\square$

# E Bandit algorithm

In this section, we present our analysis of Algorithm 1. The section is organized as follows. In Appendix E.1, we detail how we construct the confidence set. The proof that $\theta_\star$ lies in the confidence set with high probability will be presented only in Appendix E.4 so that we can continue in Appendix E.2 with the proof of the main result, Theorem 29, bounding the regret. This result differs from Theorem 11 by providing additional detail about the choice of the parameters in the algorithm. The proof presented in Appendix E.2 requires a number of technical lemmas that are presented as the proof develops. The proofs of these are postponed to subsequent sections. Before the proof of these, we devote the next section (Appendix E.3) to technical results on consequences of self-concordance which will be useful for the rest of the proofs. This is followed in Appendix E.4 by the proof that the confidence set constructed indeed has the required coverage. The next section (Appendix E.5) is devoted to proving Lemma 30 ("ellipsoidal diameter bound on the confidence set"), which is one of the two key results required for the proof of the main regret bound (besides the result on the coverage of the confidence set). A self-bounding property of self-concordance functions (Lemma 31), which is the second main ingredient of the regret bound proof is shown in Appendix E.6. Finally, for completeness, we present the (well known) elliptical potential lemma (stated here as Lemma 38) in Appendix E.7.

## E.1 Constructing the confidence set

The confidence set construction is based on first obtaining the parameter vector $\hat{\theta}_t$. This parameter vector is chosen to be the minimizer of the regularized negative log-likelihood function: $\hat{\theta}_t = \arg\min_{\theta \in \mathbb{R}^d} \mathcal{L}(\theta; \mathcal{D}_t)$ where $\mathcal{D}_t = ((X_i, Y_i))_{i=1}^{t-1}$ is the data available in step $t$ and

$$\mathcal{L}(\theta; \mathcal{D}_t, \lambda) = \frac{\lambda}{2}\|\theta\|^2 - \sum_{i=1}^{t-1} \log q(Y_i; X_i^\top \theta) \tag{32}$$

where $\lambda > 0$ is a tuning parameter (to be chosen later) and $q(y; u) = \frac{dQ_u}{dQ}(y)$ is the density of $Q_u$ with respect to $Q$ at $y \in \mathbb{R}$. It should be clear from the definitions that $q$ is well-defined. The purpose of regularization is to ensure that the loss function has a unique optimizer even in the data poor regime.

For the construction of the confidence set it will be useful to derive an equivalent expression for the loss $\mathcal{L}$. For this, first note that the density $q$ satisfies $q(y; u) = \exp(yu - \psi_Q(u))$. Plugging this into the definition of $\mathcal{L}$, we get

$$\mathcal{L}(\theta; \mathcal{D}_t, \lambda) = \frac{\lambda}{2}\|\theta\|^2 + \sum_{i=1}^{t-1}(\psi(X_i^\top \theta) - Y_i X_i^\top \theta).$$

As it is well known, $\psi$ is a convex function of its argument (Theorem 1.13 of [Bro86]) and hence $\theta \mapsto \mathcal{L}(\theta; \mathcal{D})$ is strictly convex provided that $\lambda > 0$.

For the confidence set construction we will need the non-constant part of the gradient of $\mathcal{L}(\cdot; \mathcal{D}_t)$, which we denote by $g_t$. We will also need the curvature of $\mathcal{L}(\cdot; \mathcal{D}_t)$, which we denote by $H_t$. These are

$$g_t(\theta) = \sum_{i=1}^{t-1} \mu(X_i^\top \theta) X_i + \lambda \theta \quad \text{so that} \quad \nabla_\theta \mathcal{L}(\theta; \mathcal{D}_t) = g_t(\theta) - \sum_{i=1}^{t-1} X_i Y_i\,,$$

and

$$H_t(\theta) = \nabla_\theta^2 \mathcal{L}(\theta; \mathcal{D}_t) = \lambda I + \sum_{i=1}^{t-1} \dot{\mu}(X_i^\top \theta) X_i X_i^\top\,.$$

The minimizer $\hat{\theta}_t = \arg\max_{\theta \in \mathbb{R}^d} \mathcal{L}(\theta; \mathcal{D}, \lambda)$ has the property that

$$\left. \frac{\partial \mathcal{L}(\theta; \mathcal{D}_t, \lambda)}{\partial \theta} \right|_{\theta = \hat{\theta}_t} = 0\,.$$

This implies that

$$g_t(\hat{\theta}_t) - \sum_{i=1}^{t} X_i Y_i = 0.$$

With this, we can introduce our confidence set construction, which is based on the work of [Jan+24]. For $\delta \in (0,1]$, we let

$$\mathcal{C}_t^\delta(\hat{\theta}_t) = \left\{ \theta \in \Theta \; : \; \left\| g_t(\theta) - g_t(\hat{\theta}_t) \right\|_{H_t^{-1}(\theta)} \leq \gamma_t(\delta) \right\}, \tag{33}$$

where for $M$ to be chosen later (in Lemma 28),

$$\lambda_T = 1 \vee \frac{2dM}{S_0} \log\left( e\sqrt{1 + \frac{TL}{d}} \vee 1/\delta \right), \tag{34}$$

$$\gamma_t(\delta) = \sqrt{\lambda_T}\left( \frac{1}{2M} + S_0 \right) + \frac{4Md}{\sqrt{\lambda_T}} \log\left( e\sqrt{1 + \frac{tL}{d}} \vee 1/\delta \right) \quad \text{for all} \quad t \in [T]. \tag{35}$$

Here, $S_0 = \sup\{\|\theta\| \; : \; \theta \in \Theta\}$, as defined in the main body of the paper. In the algorithm we then choose $\mathcal{C}_t$ to be $\mathcal{C}_t^\delta(\hat{\theta}_t)$ with a fixed value of $\delta \in [0,1]$ that bounds the failure probability of the algorithm. The following lemma, whose proof is postponed to Appendix E.4, as mentioned beforehand, shows that the confidence sets $\cap_{t \geq 1} \mathcal{C}_t^\delta(\hat{\theta}_t)$ have coverage $1 - \delta$:

**Lemma 28.** *Let Assumptions 1 and 2 hold and choose $M \geq \max(K/\log(2), 1/(c_1 - S_1), 1/(c_2 + S_2))$ in Eqs. (34) and (35), where $\Gamma$ is any stretch function for $(Q_u)_{u \in [S_2, S_1]}$ and $K = \sup_{S_2 \leq u \leq S_1} \Gamma(u)$. Then, for the confidence set defined in Eq. (33) and for all $\delta \in (0,1]$,*

$$\mathbb{P}(\forall t \geq 1, \theta_\star \in \mathcal{C}_t^\delta(\hat{\theta}_t)) \geq 1 - \delta.$$

Note that Theorem 7 and Assumption 2 guarantees the existence of a stretch function $\Gamma$ mentioned in the theorem.

> In the remainder of this section, we will fix $\Gamma$ to one such stretch function.

In general, here, one wants to use the smallest such stretch function (i.e., $\Gamma = \Gamma_Q$). When $\Gamma_Q$ is not available, in the lack of a better choice for $\Gamma$, the choice worked out in Theorem 7 can be used.

### E.2 Proof of regret upper bound

Let $E_\delta$ be the event that $E_\delta = \{\theta_\star \in \mathcal{C}_t^\delta(\hat{\theta}_t)\}$ which by Lemma 28 holds with probability at least $1 - \delta$. For the next theorem, recall that $S_0 = \sup_{\theta \in \Theta} \|\theta\|$ and $S_1 = \sup \mathcal{U}, S_2 = \inf \mathcal{U}$.

**Theorem 29.** *Let $\delta \in (0,1]$ and $T$ a positive integer and consider a well-posed GLB model $\mathcal{G} = (\mathcal{X}, \Theta, \mathcal{Q})$ and assume that Assumptions 1 and 2 hold. Then, by setting $\mathcal{C}_t = \mathcal{C}_t^\delta(\hat{\theta}_t)$, for any $\theta_\star \in \Theta$, with probability at least $1 - \delta$, the regret $\mathrm{Regret}(T)$ of Algorithm 1 when it interacts with the GLB instance specified by $\theta_\star$ can be upper bounded by,*

$$\mathrm{Regret}(T) \leq 8c\,\gamma_T(\delta)\sqrt{d\dot{\mu}(x_\star^\top \theta_\star)(1 + L/\lambda)\log\left(1 + \frac{LT}{d\lambda}\right)T}$$

$$+ 8c^2\,\gamma_T^2(\delta)L^2 K\kappa \log(\lambda + T/d)$$

$$+ 32c^2\,\gamma_T^2(\delta) \cdot Kd(1 + L/\lambda)\log\left(1 + \frac{LT}{d\lambda}\right),$$

*where $c = (1 + 2K(S_1 - S_2))$ and*

$$K = \sup_{S_2 \leq u \leq S_1} \Gamma(u). \tag{36}$$

*Proof.* We first consider the case that the base distribution has $0$ variance, which implies that $Q$ is a Dirac. As discussed beforehand, and as it is easy to see it, in this case $\mathcal{U}_Q = \mathbb{R}$, $Q_u = Q$ for any $u \in \mathbb{R}$. Hence, all arms have the same payoff and all algorithm incur zero regret. In the rest of this proof, we assume that $\mathrm{Var}(Q) > 0$. Since $\mu(\cdot) = \dot{\psi}_Q(\cdot)$ is infinitely differentiable (Proposition 2), we can perform a second-order Taylor expansion on the regret

$$\mathrm{Regret}(T) = \sum_{t=1}^{T} \mu(x_\star^\top \theta_\star) - \mu(X_t^\top \theta_\star)$$

$$= \underbrace{\sum_{t=1}^{T} \dot{\mu}(X_t^\top \theta_\star)(x_\star - X_t)^\top \theta_\star}_{R_1(T)} + \underbrace{\frac{1}{2} \sum_{t=1}^{T} \ddot{\mu}(\xi_t)((x_\star - X_t)^\top \theta_\star)^2}_{R_2(T)},$$

where $\xi_t$ is between $X_t^\top \theta_\star$ and $x_\star^\top \theta_\star$ for all $t \in [T]$. On event $E_\delta$, by definition of $X_t, \theta_t$ (in Algorithm 1), it holds that $x_\star^\top \theta_\star \leq X_t^\top \theta_t$. Observe that $\gamma_t(\delta)$ (Eq. (35)) is increasing in $t$, we have that $\gamma_t(\delta) \leq \gamma_T(\delta)$ for all $t \in [T]$. Then we can bound $R_1(T)$ as follows

$$R_1(T) = \sum_{t=1}^{T} \dot{\mu}(X_t^\top \theta_\star)(x_\star - X_t)^\top \theta_\star$$

$$\leq \sum_{t=1}^{T} \dot{\mu}(X_t^\top \theta_\star) X_t^\top (\theta_t - \theta_\star)$$

$$\leq \sum_{t=1}^{T} \dot{\mu}(X_t^\top \theta_\star) \|X_t\|_{H_t^{-1}(\theta_\star)} \|\theta_t - \theta_\star\|_{H_t(\theta_\star)}$$

where the last inequality is due to Cauchy-Schwarz. Since $\theta_t$ and $\theta_\star$ are all in the confidence set on $E_\delta$, we are able to bound $\|\theta_t - \theta_\star\|_{H_t(\theta_\star)}$ by the following lemma that exploits the properties of confidence set as well as self-concordant functions. This lemma is a variation of proposition 4 of Abeille, Faury, and Calauzènes [AFC21] where they show the result for logistic function that is 1-self-concordant.

**Lemma 30** ($\mathcal{C}_t^\delta(\hat{\theta}_t)$ has small ellipsoidal diameters)**.** *Under Assumptions 1 and 2, for all $\theta_1, \theta_2 \in \mathcal{C}_t^\delta(\hat{\theta}_t)$, it follows that*

$$\|\theta_1 - \theta_2\|_{H_t(\theta_1)} \vee \|\theta_1 - \theta_2\|_{H_t(\theta_2)} \leq 2(1 + 2K \cdot (S_1 - S_2))\gamma_t(\delta),$$

*where $K$ is defined in Eq. (36).*

By Lemma 30, we can upper bound $R_1(T)$ to be

$$R_1(T) \leq \sum_{t=1}^{T} \dot{\mu}(X_t^\top \theta_\star) \|X_t\|_{H_t^{-1}(\theta_\star)} \cdot 2(1 + 2K(S_1 - S_2))\gamma_T(\delta).$$

Denote $A_t = \sqrt{\dot{\mu}(X_t^\top \theta_\star)} X_t$ and we have that $H_t(\theta_\star) = \sum_{s=1}^{t} A_t A_t^\top + \lambda I$ as well as $\|A_t\| \leq \sqrt{L} \leq L$ where the second inequality is because WLOG we assume $L \geq 1$ in Assumption 2. We can

bound $R_1(T)$ in the terms of $A_t$.

$$R_1(T) \le 2(1 + 2K(S_1 - S_2))\gamma_T(\delta) \sum_{t=1}^{T} \sqrt{\dot{\mu}(X_t^\top \theta_\star) \|X_t\|_{H_t^{-1}(\theta_\star)}^2} \sqrt{\dot{\mu}(X_t^\top \theta_\star)}$$

$$\le 2(1 + 2K(S_1 - S_2))\gamma_T(\delta) \sqrt{\sum_{t=1}^{T} \dot{\mu}(X_t^\top \theta_\star) \|X_t\|_{H_t^{-1}(\theta_\star)}^2} \sqrt{\sum_{t=1}^{T} \dot{\mu}(X_t^\top \theta_\star)}$$

$$\text{(Cauchy-Schwarz)}$$

$$= 2(1 + 2K(S_1 - S_2))\gamma_T(\delta) \sqrt{\sum_{t=1}^{T} \|A_t\|_{H_t^{-1}(\theta_\star)}^2} \sqrt{\sum_{t=1}^{T} \dot{\mu}(X_t^\top \theta_\star)}$$

$$\le 4(1 + 2K(S_1 - S_2))\gamma_T(\delta) \sqrt{d(1 + L/\lambda) \log\left(1 + \frac{LT}{d\lambda}\right)} \sqrt{\sum_{t=1}^{T} \dot{\mu}(X_t^\top \theta_\star)}$$

where in the last step we use elliptical potential lemma of Abbasi-Yadkori, Pál, and Szepesvári [APS11], which, for easy of reference, we give in Lemma 38. We now start to bound $R_2(T)$. For convenience, we throw away the factor of $1/2$.

$$R_2(T) \le \sum_{t=1}^{T} \ddot{\mu}(\xi_t)((x_\star - X_t)^\top \theta_\star)^2$$

$$\le \sum_{t=1}^{T} \ddot{\mu}(\xi_t)(X_t^\top(\theta_t - \theta_\star))^2 \qquad (X_t^\top \theta_t \ge x_\star^\top \theta_\star \ge X_t^\top \theta_\star)$$

$$\le \sum_{t=1}^{T} \ddot{\mu}(\xi_t) \|X_t\|_{H_t^{-1}(\theta_\star)}^2 \|\theta_t - \theta_\star\|_{H_t(\theta_\star)}^2$$

$$\le 4(1 + 2K(S_1 - S_2))^2 \gamma_T(\delta)^2 \sum_{t=1}^{T} \ddot{\mu}(\xi_t) \|X_t\|_{H_t^{-1}(\theta_\star)}^2$$

where in the last inequality we use Lemma 30. By definition of self-concordant function, we have that $\ddot{\mu}(\xi_t) \le \Gamma(\xi_t)\dot{\mu}(\xi_t)$. Since $\xi_t$ is between $x_\star^\top \theta_\star$ and $X_t^\top \theta_\star$. Note that $\Gamma$ defined in Theorem 7 is increasing on $[0, c_1)$ and decreasing on $(-c_2, 0)$, which gives us $\Gamma(\xi_t) \le \Gamma(X_t^\top \theta_\star) \vee \Gamma(x_\star^\top \theta_\star) \le K$. Let $V_t = \lambda I + \sum_{i=1}^{t} X_i X_i^\top$. We hence have

$$R_2(T) \le 4(1 + 2K(S_1 - S_2))^2 \gamma_T(\delta)^2 \sum_{t=1}^{T} K\dot{\mu}(\xi_t) \|X_t\|_{H_t^{-1}(\theta_\star)}^2$$

$$\le 4(1 + 2K(S_1 - S_2))^2 \gamma_T(\delta)^2 KL \sum_{t=1}^{T} \|X_t\|_{H_t^{-1}(\theta_\star)}^2 \qquad (\dot{\mu}(\cdot) \le L \text{ (Assumption 2)})$$

$$\le 4(1 + 2K(S_1 - S_2))^2 \gamma_T(\delta)^2 KL\kappa \cdot \sum_{t=1}^{T} \|X_t\|_{V_t^{-1}}^2 \qquad (H_t^{-1}(\theta_\star) \preceq \kappa V_t^{-1})$$

$$\le 4(1 + 2K(S_1 - S_2))^2 \gamma_T(\delta)^2 KL\kappa \cdot L \log(\lambda + T/d). \qquad (\text{Lemma 38})$$

Putting the bound on $R_1(T)$ and $R_2(T)$ together, we have that

$$\text{Regret}(T) = R_1(T) + R_2(T)$$

$$\le 4(1 + 2K(S_1 - S_2))\gamma_T(\delta) \sqrt{d(1 + L/\lambda) \log\left(1 + \frac{LT}{d\lambda}\right)} \sqrt{\sum_{t=1}^{T} \dot{\mu}(X_t^\top \theta_\star)}$$

$$+ 4(1 + 2K(S_1 - S_2))^2 \gamma_T(\delta)^2 L^2 K\kappa \log(\lambda + T/d). \qquad (37)$$

We mimic the trick used in Janz, Liu, Ayoub, and Szepesvári [Jan+24] to bound $\sqrt{\sum_{t=1}^{T} \dot{\mu}(X_t^\top \theta_\star)}$ which was originally proposed by Abeille, Faury, and Calauzènes [AFC21]. We present the following lemma that is abstracted out from Claim 14 of Janz, Liu, Ayoub, and Szepesvári [Jan+24].

**Lemma 31** (Self-bounding property of self-concordance functions). *Let $\mathcal{V} = [a, b]$, a closed, nonempty interval over the reals, $f$ a real valued function defined over an interval of the reals that is twice continuously differentiable over $\mathcal{V}$ such that for some $\Gamma : \mathcal{V} \to \mathbb{R}_+$, $|\ddot{f}(v)| \leq \Gamma(v)\dot{f}(v)$ holds for all $v \in \mathcal{V}$. Assume that $A = \sup_{v \in \mathcal{V}} \Gamma(v) < \infty$. Furthermore, assume that either $\dot{f}$ is identically zero over $\mathcal{V}$, or $\dot{f}$ is positive valued over $\mathcal{V}$. For $n$ a positive integer, let $\{a_t\}_{t=1}^n \subset \mathcal{V}$. Then,*

$$\sum_{t=1}^n \dot{f}(a_t) \leq n\dot{f}(b) + A\sum_{t=1}^n f(b) - f(a_t).$$

We apply this lemma with $f = \mu$, $[a, b] = [S_2, x_\star^\top \theta_\star] \subset [S_2, S_1]$ and $\Gamma$ restricted to $[a, b]$. Then, all the conditions of the lemma are satisfied by our choice of $\Gamma$. Furthermore, $A = \sup_{v \in [S_2, x_\star^\top \theta_\star]} \Gamma(v) \leq K < +\infty$. Hence, all the conditions of the lemma are verified. Hence,

$$\sqrt{\sum_{t=1}^T \dot{\mu}(X_t^\top \theta_\star)} \leq \sqrt{T\dot{\mu}(x_\star^\top \theta_\star) + K\mathrm{Regret}(T)}$$

$$\leq \sqrt{T\dot{\mu}(x_\star^\top \theta_\star)} + \sqrt{K\mathrm{Regret}(T)}. \tag{38}$$

Plug Eq. (38) into Eq. (37),

$$\mathrm{Regret}(T) \leq 4(1 + 2K(S_1 - S_2))\gamma_T(\delta)\sqrt{d\dot{\mu}(x_\star^\top \theta_\star)(1 + L/\lambda)\log\left(1 + \frac{LT}{d\lambda}\right)T}$$

$$+ 4(1 + 2K(S_1 - S_2))^2\gamma_T(\delta)^2 L^2 K\kappa \log(\lambda + T/d)$$

$$+ 4(1 + 2K(S_1 - S_2))\gamma_T(\delta)\sqrt{Kd(1 + L/\lambda)\log\left(1 + \frac{LT}{d\lambda}\right)}\sqrt{\mathrm{Regret}(T)}.$$

Let

$$A = 4(1 + 2K(S_1 - S_2))\gamma_T(\delta)\sqrt{Kd(1 + L/\lambda)\log\left(1 + \frac{LT}{d\lambda}\right)},$$

$$B = 4(1 + 2K(S_1 - S_2))\gamma_T(\delta)\sqrt{d\dot{\mu}(x_\star^\top \theta_\star)(1 + L/\lambda)\log\left(1 + \frac{LT}{d\lambda}\right)T}$$

$$+ 4(1 + 2K(S_1 - S_2))^2\gamma_T(\delta)^2 L^2 K\kappa \log(\lambda + T/d),$$

we can write out the inequality

$$\mathrm{Regret}(T) \leq A\sqrt{\mathrm{Regret}(T)} + B.$$

Solving it we have that

$$\mathrm{Regret}(T) \leq 2A^2 + 2B.$$

Plugging in the definition of $A$ and $B$ back,

$$\mathrm{Regret}(T) \leq 8(1 + 2K(S_1 - S_2))\gamma_T(\delta)\sqrt{d\dot{\mu}(x_\star^\top \theta_\star)(1 + L/\lambda)\log\left(1 + \frac{LT}{d\lambda}\right)T}$$

$$+ 8(1 + 2K(S_1 - S_2))^2\gamma_T(\delta)^2 L^2 K\kappa \log(\lambda + T/d)$$

$$+ 32(1 + 2K(S_1 - S_2))^2\gamma_T(\delta)^2 \cdot Kd(1 + L/\lambda)\log\left(1 + \frac{LT}{d\lambda}\right). \qquad \square$$

### E.3 Self-concordance control

In this section we provide technical results about self-concordant functions which play important roles in confidence set construction and controlling the regret of Algorithm 1. Specifically, Corollary 33 and Lemma 34 are used to show Lemma 30, one of the key lemmas we use in the proof of Theorem 29. Lemma 5 is used to justify the confidence set contains $\theta_\star$ with high probability (Lemma 28).

The next lemma shows that for self-concordant NEFs, $\dot{\mu}$ is a smooth function of its argument. The lemma is essentially the same as Lemma 3 from Janz, Liu, Ayoub, and Szepesvári [Jan+24] (itself based on a result of Sun and Tran-Dinh [ST19]) and is updated only to match our definitions of self-concordance, which is a refinement of that used by Janz, Liu, Ayoub, and Szepesvári [Jan+24]. The proof (based on the proof of a similar result of Sun and Tran-Dinh [ST19]) is included for the convenience of the reader.

**Lemma 32** (Self-concordance to smoothness). *Let $\mathcal{U}$ be an interval over the reals, $\mu$ a real valued function defined over an interval of the reals that is twice continuously differentiable over $\mathcal{U}$ such that for some $\Gamma : \mathcal{U} \to \mathbb{R}_+$, $|\ddot{\mu}(u)| \leq \Gamma(u)\dot{\mu}(u)$ holds for all $u \in \mathcal{U}$. Assume that $K = \sup_{u \in \mathcal{U}} \Gamma(u) < \infty$. Assume that either $\dot{\mu}$ is identically zero over $\mathcal{U}$, or $\dot{\mu}$ is positive valued over $\mathcal{U}$. Then, for any $u, u' \in \mathcal{U}$,*

$$\dot{\mu}(u') \leq \dot{\mu}(u)e^{K|u-u'|}.$$

An immediate corollary of this lemma is that self-concordance of a NEF implies that the variance function, $\dot{\mu}$, of the NEF is smooth:

**Corollary 33** (Self-concordance to smoothness). *Let $\mathcal{Q} = (Q_u)_{u \in \mathcal{U}}$ be self-concordant with stretch function $\Gamma : \mathcal{U} \to \mathbb{R}_+$, where $\mathcal{U}$ is an interval and assume $K = \sup_{u \in \mathcal{U}} \Gamma(u) < \infty$. Then for any $u, u' \in \mathcal{U}$,*

$$\dot{\mu}(u') \leq \dot{\mu}(u)e^{K|u-u'|}.$$

Note that the inequality is well-posed since $u, u' \in \mathcal{U}_Q^\circ$, and $\mu$ is known to be differentiable over $\mathcal{U}_Q^\circ$ and, by the definition of self-concordance, $\mathcal{U} \subset \mathcal{U}_Q^\circ$.

*Proof.* This result follows from Lemma 32 once we notice that the variance function of a NEF is such that if $\dot{\mu}(u) = 0$ for any $u \in \mathcal{U}$, then $\dot{\mu}$ is identically zero over $\mathcal{U}$. Indeed, if $\dot{\mu}(u) = 0$, then $Q_u$ is a Dirac distribution and so is $Q_v$ for any $v \in \mathcal{U}$. $\qquad\square$

*Proof of Lemma 32.* When $\dot{\mu}$ is identically zero over $\mathcal{U}$, the statement is trivial. Hence, consider now the case when $\dot{\mu}$ is positive valued over $\mathcal{U}$:

$$\dot{\mu}(v) > 0 \qquad \text{for all} \quad v \in \mathcal{U}. \tag{39}$$

Then, it suffices to show that $\ln \frac{\dot{\mu}(u')}{\dot{\mu}(u)} \leq K|u - u'|$. To show this, define $\phi(t) = \dot{\mu}(u + t(u' - u))$ so that $\phi(0) = \dot{\mu}(u)$ and $\phi(1) = \dot{\mu}(u')$. Since $\mathcal{U}_Q$ is an interval with non-empty interior, $\phi(t)$ is well-defined for all $t \in [0, 1]$. Furthermore, by Eq. (39) and since $\mathcal{U}$ is an interval, we have that $\phi(t) > 0$ for all $t \in [0, 1]$. Consider now the map $t \mapsto \ln \phi(t)$ where $t \in [0, 1]$. The derivative of this map exist and is continuous over $(0, 1)$, and in particular, $\frac{d}{dt} \ln \phi(t) = \frac{\dot{\phi}(t)}{\phi(t)}$ by the chain rule. Indeed, the derivative of $\phi$ exists and is continuous over $(0, 1)$, because the same holds for $\dot{\mu}$ by the properties of NEFs, and as we just discussed, $\phi$ is positive over $[0, 1]$ and is continuous. Now, by the fundamental theorem of calculus applied to $t \mapsto \frac{d}{dt} \ln \phi(t)$ and by the monotonicity of integrals,

$$\ln \frac{\dot{\mu}(u')}{\dot{\mu}(u)} = \ln \frac{\phi(1)}{\phi(0)} = \int_0^1 \frac{d \ln \phi(t)}{dt} dt \leq \int_0^1 \left| \frac{d \ln \phi(t)}{dt} \right| dt. \tag{40}$$

It remains to bound the integrand in the rightmost expression. For this, as discussed earlier we have

$$\left| \frac{d \ln \phi(t)}{dt} \right| = \left| \frac{\phi'(t)}{\phi(t)} \right| = \frac{|\phi'(t)|}{\phi(t)}. \tag{41}$$

To bound the ratio on the right, we again use the chain rule and calculate

$$|\phi'(t)| = |\ddot{\mu}(u + t(u' - u))||u' - u| \leq K \underbrace{\dot{\mu}(u + t(u' - u))}_{\phi(t)} |u' - u|,$$

where the inequality follows by the definition of $K$ by definition of self-concordant function, we have that for all $u \in \mathcal{U}$, $|\ddot{\mu}(u)| \leq K\dot{\mu}(u)$. Now, the result follows since we have shown that the integrand is upper bounded by $K|u - u'|$ and thus $\int_0^1 \left| \frac{d \ln \phi(t)}{dt} \right| dt \leq K|u - u'|$. $\qquad\square$

We continue with two results, both of which use the lemma just proved. The first result gives a lower bound for the integral remainder term when Taylor's theorem is used to approximate $\mu$. The second result gives a quadratic upper bound on the CGF of $Q_u$, and will be the basis for constructing our confidence set.

**Lemma 34.** *Let $\mathcal{Q} = (Q_u)_{u \in \mathcal{U}}$ be self-concordant with stretch function $\Gamma : \mathcal{U} \to \mathbb{R}_+$ where $\mathcal{U}$ is an interval, and assume $K = \sup_{u \in \mathcal{U}} \Gamma(u) < \infty$. Then for any $u, u' \in \mathcal{U}$,*

$$\int_0^1 \dot{\mu}(u + t(u' - u)) dt \geq \frac{\dot{\mu}(u)}{1 + K|u - u'|}.$$

*Proof.* By Corollary 33, it follows that

$$\dot{\mu}(u + t(u' - u)) \geq \dot{\mu}(u) \exp(-Kt|u' - u|).$$

Integrating both sides between 0 and 1 gives us

$$\begin{aligned}
\int_0^1 \dot{\mu}(u + t(u' - u)) &\geq \dot{\mu}(u) \int_0^1 \exp(-Kt|u' - u|) dv \\
&= \dot{\mu}(u) \left[ \frac{-\exp(-Kt|u' - u|)}{K|u' - u|} \right]_0^1 \\
&= \dot{\mu}(u) \frac{1 - \exp(-K|u' - u|)}{K|u' - u|} \\
&\geq \frac{\dot{\mu}(u)}{1 + K|u' - u|},
\end{aligned}$$

where the last inequality follows from the elementary inequality $(1 - e^{-x})/x \geq 1/(1 + x)$ that holds for all $x \geq 0$. $\qquad\square$

We are now ready to prove Lemma 5. As noted beforehand, we adopt this lemma from the work of Janz, Liu, Ayoub, and Szepesvári [Jan+24]. In particular, it is an adaptation of their Lemma 1, which was proved for distributions where $\mathcal{U}_Q = \mathbb{R}$. Here, we deal with the case when $\mathcal{U}_Q$ is possibly a strict subset of $\mathbb{R}$.

**Lemma 5** (From self-concordance to light tails). *Let $\mathcal{Q} = (Q_u)_{u \in \mathcal{U}}$ be a NEF which is self-concordant with stretch function $\Gamma : \mathcal{U} \to \mathbb{R}_+$ where $\mathcal{U}$ is a subinterval of $\mathcal{U}_Q^\circ = (a, b)$. Then, for any $u \in \mathcal{U}$,*

$$\psi_{Q_u}(s) \leq s\mu(u) + s^2 \dot{\mu}(u) \quad \text{for all} \quad s \in [-\log(2)/K, \log(2)/K] \cap (a - \inf \mathcal{U}, b - \sup \mathcal{U}), \quad (2)$$

*where $K = \sup_{u \in \mathcal{U}} \Gamma(u)$.*

*Proof.* Let $u \in \mathcal{U}$. Hence, by our assumption on $\mathcal{U}$, $u \in \mathcal{U}_Q$. Now let $s \in \mathbb{R}$. Then,

$$\begin{aligned}
\psi_{Q_u}(s) &= \log \int \exp(sy) Q_u(dy) = \log \left[ \frac{1}{M_Q(u)} \int \exp(sy) \exp(uy) Q(dy) \right] \\
&= \psi_Q(u + s) - \psi_Q(u).
\end{aligned}$$

Hence, $\psi_{Q_u}(s)$ is finite valued whenever $u + s \in \mathcal{U}_Q$. Assume that this holds and in fact $u + s \in \mathcal{U}_Q^\circ$.

Since $u, u + s \in \mathcal{U}_Q^\circ$, $\psi_Q$ is twice continuously differentiable over an open interval containing $u$ and $u + s$. Then, by Taylor's theorem there exists $\xi$ in the closed interval between $u$ and $u + s$ such that

$$\psi_Q(u + s) - \psi_Q(u) = s\dot{\psi}_Q(u) + \frac{s^2}{2} \ddot{\psi}_Q(\xi).$$

Since $\dot{\psi}_Q = \mu$ and $\ddot{\psi}_Q = \dot{\mu}$ (cf. Proposition 2) we get

$$\psi_Q(u + s) - \psi_Q(u) = s\mu(u) + \frac{s^2}{2} \dot{\mu}(\xi).$$

Now, by Corollary 33, we have that

$$\dot{\mu}(\xi) \leq \dot{\mu}(u) \cdot e^{K|u-\xi|} \leq \dot{\mu}(u)e^{Ks} \leq 2\dot{\mu}(u),$$

where the final inequality follows when $|s| \leq \log(2)/K$. Putting things together, it follows that if $|s| \leq \log(2)/K$ and $s \in \mathcal{U}_Q^\circ - \{u\}$ then

$$\psi_{Q_u}(s) = \psi_Q(u+s) - \psi_Q(u) \leq s\mu(u) + s^2\dot{\mu}(u).$$

For $S \subset \mathbb{R}$, $r \in \mathbb{R}$, let $S \pm r = \{s \pm r : s \in S\}$. Since $u$ is an arbitrary point in $\mathcal{U}$, the above conditions on $s$ will be satisfied if $|s| \leq \log(2)/K$ and $s \in Z \doteq \cap_{u \in \mathcal{U}} \mathcal{U}_Q^\circ - u$. Now, from $\mathcal{U}_Q^\circ = (a, b)$, we have $Z = \cap_{u \in \mathcal{U}}(a - u, b - u) = (\sup_{u \in \mathcal{U}} a - u, \inf_{u \in \mathcal{U}} b - u) = (a - \inf \mathcal{U}, b - \sup \mathcal{U})$, finishing the proof. $\qquad\square$

From the calculation at the end of the proof it follows that the statement of the lemma is non-vacuous if for $\mathcal{U}_Q^\circ = (a, b)$, $a - \inf \mathcal{U} < 0 < b - \sup \mathcal{U}$, which is equivalent to that $a < \inf \mathcal{U}$ and $\sup \mathcal{U} < b$, which is always satisfied when $\mathcal{U}$ is a strict subset of $\mathcal{U}_Q^\circ$.

### E.4 Confidence set construction

We now turn to proving Lemma 28 which is concerned with showing that the confidence sets $\mathcal{C}_t^\delta$ contain the true parameter $\theta_\star$ with probability $1 - \delta$. The proof is essentially the same as that of Lemma 4 of [Jan+24]. We will need the following result, which is taken verbatim from the paper of Janz, Liu, Ayoub, and Szepesvári [Jan+24].

**Proposition 35** (Theorem 2 of [Jan+24]). *Fix $\lambda, M > 0$. Let $(X_t)_{t \in \mathbb{N}^+}$ be a $B_2^d$-valued random sequence, $(Y_t)_{t \in \mathbb{N}^+}$ a real valued random sequence and $(\nu_t)_{t \in \mathbb{N}}$ be a nonnegative valued random sequence. Let $\mathbb{F} = (\mathbb{F}_t)_{t \in \mathbb{N}}$ be a filtration such that (i) $(X_t)_{t \in \mathbb{N}^+}$ is $\mathbb{F}$-predictable and (ii) $(Y_t)_{t \in \mathbb{N}^+}$ are $\mathbb{F}$-adapted. Let $\epsilon_t = Y_t - \mathbb{E}[Y_t | \mathbb{F}_{t-1}]$ and assume that the following condition holds:*

$$\mathbb{E}[\exp(s\epsilon_t)|\mathbb{F}_{t-1}] \leq \exp(s^2 \nu_{t-1}) \quad \text{for all} \quad |s| \leq 1/M \text{ and } t \in \mathbb{N}^+.$$

*Then, for $\tilde{H}_t = \sum_{i=1}^t \nu_{i-1} X_i X_i^\top + \lambda I$ and $\mathbf{S}_t = \sum_{i=1}^t \epsilon_i X_i$ and $\delta > 0$,*

$$\mathbb{P}\left(\exists t \in \mathbb{N}^+ : \|\mathbf{S}_t\|_{\tilde{H}_t^{-1}} \geq \frac{\sqrt{\lambda}}{2M} + \frac{2M}{\sqrt{\lambda}} \log\left(\frac{\det(\tilde{H}_t)^{1/2}\lambda^{-d/2}}{\delta}\right) + \frac{2M}{\sqrt{\lambda}} d\log(2)\right) \leq \delta.$$

We now turn to proving Lemma 28. For the convenience of the reader, we start by recalling the definition of the confidence sets $\mathcal{C}_t^\delta$ involved (cf. Eq. (33)). Recall that $c_2 < S_2 \leq x^\top \theta \leq S_1 < c_1$ for $x \in \mathcal{X}$ and $\theta \in \Theta$. For $\delta \in (0, 1]$, we have

$$\mathcal{C}_t^{\delta,\lambda}(\hat{\theta}_t) = \left\{\theta \in \Theta : \left\|g_t(\theta) - g_t(\hat{\theta}_t)\right\|_{H_t^{-1}(\theta)} \leq \gamma_t(\delta)\right\}$$

where

$$\gamma_t(\delta) = \sqrt{\lambda_T}\left(\frac{1}{2M} + S_0\right) + \frac{4Md}{\sqrt{\lambda_T}} \log\left(e\sqrt{1 + \frac{tL}{d}} \vee 1/\delta\right) \quad \text{for all} \quad t \in [T],$$

$$\lambda_T = 1 \vee \frac{2dM}{S_0} \log\left(e\sqrt{1 + \frac{TL}{d}} \vee 1/\delta\right),$$

and recall that $S_0 = \sup_{\theta \in \Theta} \|\theta\|$ or an upper bound on this quantity, and $M$ is specified in the next result:

**Lemma 28.** *Let Assumptions 1 and 2 hold and choose $M \geq \max(K/\log(2), 1/(c_1 - S_1), 1/(c_2 + S_2))$ in Eqs. (34) and (35), where $\Gamma$ is any stretch function for $(Q_u)_{u \in [S_2, S_1]}$ and $K = \sup_{S_2 \leq u \leq S_1} \Gamma(u)$. Then, for the confidence set defined in Eq. (33) and for all $\delta \in (0, 1]$,*

$$\mathbb{P}(\forall t \geq 1, \theta_\star \in \mathcal{C}_t^\delta(\hat{\theta}_t)) \geq 1 - \delta.$$

*Proof.* From definition it follows that $\theta_\star \in \Theta$. Now we prove that with probability at least $1 - \delta$, it holds that $\|g_t(\theta_\star) - g_t(\hat{\theta}_t)\| \leq \gamma_T(\delta)$ for all $t \geq 1$. The proof goes through by using Proposition 35 and we now match the conditions of Proposition 35. By Assumption 1, we have that $(X_t)_{t\in\mathbb{N}^+}$ is a $B_2^d$-valued random sequence. Let $\mathcal{F}_{t-1} = \sigma(X_1, Y_1, ..., X_{t-1}, Y_{t-1}, X_t)$ for $t \geq 1$. Consider the filtration $\mathcal{F} = (\mathcal{F}_t)_{t\in\mathbb{N}}$. Then by definition, $(X_t)_{t\in\mathbb{N}^+}$ are $\mathcal{F}$-predictable and $(Y_t)_{t\in\mathbb{N}^+}$ are $\mathcal{F}$-adapted. Note that $\mu(X_i^\top \theta_\star) = \mathbb{E}[Y_i|\mathcal{F}_{i-1}]$ for all $i \in [n]$. Let $\varepsilon_i = Y_i - \mathbb{E}[Y_i|\mathcal{F}_{i-1}]$. This gives $(\varepsilon_t)_{t\in\mathbb{N}^+}$ are also $\mathcal{F}$-adapted and the following identity follows by definition

$$g_t(\hat{\theta}_t) - g_t(\theta_\star) = \sum_{i=1}^{t} \varepsilon_i X_i + \lambda \theta_\star$$
$$= \mathbf{S}_t + \lambda \theta_\star,$$

where $\mathbf{S}_t = \sum_{i=1}^{t} \varepsilon_i X_i$. Let

$$\nu_{t-1} = \dot{\mu}(X_t^\top \theta_\star).$$

Now we would like to apply Lemma 5 to show that, for $|s| \leq M$, it follows that

$$\mathbb{E}[\exp(s\varepsilon_t)|\mathcal{F}_{t-1}] \leq \exp(s^2 \nu_{t-1}). \tag{42}$$

Applying the definition of $\varepsilon_t$, it follows that

$$\mathbb{E}[\exp(s\varepsilon_t)|\mathcal{F}_{t-1}] = \mathbb{E}[\exp(sY_t - s\mu(X_t^\top \theta_\star)|\mathcal{F}_{t-1}]$$
$$= \exp(-s\mu(X_t^\top \theta_\star))\mathbb{E}[Y_t|\mathcal{F}_{t-1}] \quad \text{a.s.},$$

where the last equality is because $X_t$ is $\mathcal{F}_{t-1}$-measurable. Since, by definition, the distribution of $Y_t$ given $\mathcal{F}_{t-1}$ is $Q_{X_t^\top \theta_\star}$, we have that

$$\mathbb{E}[\exp(s\varepsilon_t)|\mathcal{F}_{t-1}] = \exp(-s\mu(X_t^\top \theta_\star))\mathbb{E}[\exp(sY_t)|\mathcal{F}_{t-1}]$$
$$= \exp(-s\mu(X_t^\top \theta_\star)) \int_\mathbb{R} e^{sy} Q_{X_t^\top \theta_\star}(dy)$$
$$= \exp(-s\mu(X_t^\top \theta_\star) + \psi_{Q_{X_t^\top \theta_\star}}(s))$$
$$\leq \exp(s^2 \dot{\mu}(X_t^\top \theta_\star)),$$

where the last inequality is because $|s| \leq M$ implies that $s \in [-\log 2/K, \log 2/K] \cap (\mathcal{U}_Q^\circ - S_2) \cap (\mathcal{U}_Q^\circ - S_1)$ so Lemma 5 is applicable (it is applied with $(Q_u)_{u\in[S_2, S_1]}$ and $\Gamma$ as chosen in the statement). Then, Eq. (42) follows by noting that $\nu_{t-1} = \dot{\mu}(X_t^\top \theta_\star)$.

Lastly as defined above, $\tilde{H}_t$ corresponds to $H_t(\theta_\star)$. Taking the $\ell_2$-norm weighted by $H_t^{-1}(\theta_\star)$ and applying triangle inequality,

$$\|g_t(\hat{\theta}_t) - g_t(\theta_\star)\|_{H_t^{-1}(\theta_\star)} \leq \|\mathbf{S}_t\|_{H_t^{-1}(\theta_\star)} + \lambda\|\theta_\star\|_{H_t^{-1}(\theta_\star)} \leq \|\mathbf{S}_t\|_{H_t^{-1}(\theta_\star)} + \sqrt{\lambda}S_0,$$

where the last inequality follows by $H_t^{-1}(\theta_\star) \preceq \lambda^{-1}I$. By Proposition 35, with probability at least $1 - \delta$, it follows that for all $t \geq 1$,

$$\|\mathbf{S}_t\|_{H_t^{-1}(\theta_\star)} < \frac{\sqrt{\lambda}}{2M} + \frac{2M}{\sqrt{\lambda}} \log\left(\frac{\det(H_t(\theta_\star))^{1/2}/\lambda^{d/2}}{\delta}\right) + \frac{2M}{\sqrt{\lambda}} d\log(2).$$

We now bound $\det(H_t(\theta_\star))/\lambda^d$. Let $A_i = \sqrt{\dot{\mu}(X_i^\top \theta_\star)} X_i$ for all $i \in [t]$, then $H_t(\theta_\star)$ can be written as

$$H_t(\theta_\star) = \lambda I + \sum_{s=1}^{t} A_i A_i^\top.$$

By Assumption 2, it holds that $\sqrt{\dot{\mu}(X_t^\top \theta_\star)} \leq \sqrt{L}$, thus $\|A_i\|_2 \leq \sqrt{L} \leq L$ for all $i \in [t]$. Eq. (20.9) (Note 1 of section 20.2) in [LS20] gives

$$\det(H_t(\theta_\star))/\lambda^d \leq \left(1 + \frac{tL}{\lambda d}\right)^d.$$

The stated result follows by chaining all the inequalities together and noting that

$$\gamma_t(\delta) \geq \sqrt{\lambda}\left(\frac{1}{2M} + S_0\right) + \frac{2Md}{\sqrt{\lambda}}\left(1 + \frac{1}{2}\log\left(1 + \frac{tL}{\lambda d}\right)\right) + \frac{2M}{\sqrt{\lambda}} \log(1/\delta), \quad \text{for all} \quad t \in [T]. \tag{43}$$

$\square$

## E.5 Proof of Lemma 30

The following two lemmas (Lemmas 36 and 37) are variations of Claim 4 and Claim 3 of Janz, Liu, Ayoub, and Szepesvári [Jan+24]. The difference is that Janz, Liu, Ayoub, and Szepesvári [Jan+24] show them for all $\theta_1, \theta_2 \in \mathbb{R}^d$ because the MGF $M_Q$ therein is finite on $\mathbb{R}$. In our setting, there could be $x \in \mathcal{X}$ for some $\theta \notin \Theta$ such that $M_Q(x^\top \theta) = \infty$, hence we show it within the parameter set $\Theta$.

**Lemma 36.** *For all $\theta_1, \theta_2 \in \Theta$, it follows that*

$$g_t(\theta_1) - g_t(\theta_2) = G_t(\theta_1, \theta_2)(\theta_1 - \theta_2).$$

*In particular, we have that*

$$\|g_t(\theta_1) - g_t(\theta_2)\|_{G_t^{-1}(\theta_1, \theta_2)} = \|\theta_1 - \theta_2\|_{G_t(\theta_1, \theta_2)}.$$

*Proof.* The "In particular" part follows from definition of $\ell_2$-norm weighted by $G_t^{-1}(\theta_1, \theta_2)$. We now prove $g_t(\theta_1) - g_t(\theta_2) = G_t(\theta_1, \theta_2)(\theta_1 - \theta_2)$. By definition of the difference quotient $\alpha(\cdot, \cdot)$, we have that

$$\mu(u) - \mu(u') = \alpha(u, u')(u - u'). \tag{44}$$

Writing out the expression of $g_t(\theta_1) - g_t(\theta_2)$ gives

$$
\begin{aligned}
g_t(\theta_1) - g_t(\theta_2) &= \sum_{i=1}^{t} \left( \mu(X_i^\top \theta_1) - \mu(X_i^\top \theta_2) \right) X_i + \lambda(\theta_1 - \theta_2) \\
&= \sum_{i=1}^{t} \left( \alpha(X_i^\top \theta_1, X_i^\top \theta_2) X_i^\top (\theta_1 - \theta_2) \right) X_i + \lambda(\theta_1 - \theta_2) \qquad \text{(Eq. (44))} \\
&= \left( \sum_{i=1}^{t} \alpha(X_i^\top \theta_1, X_i^\top \theta_2) X_i X_i^\top \right) (\theta_1 - \theta_2) + \lambda(\theta_1 - \theta_2) \\
&= G_t(\theta_1, \theta_2)(\theta_1 - \theta_2). \qquad \qquad \square
\end{aligned}
$$

**Lemma 37.** *Under Assumptions 1 and 2, for all $\theta_1, \theta_2 \in \Theta$, it follows that*

$$G_t(\theta_1, \theta_2) \succeq (1 + 2K \cdot (S_1 - S_2))^{-1} H_t(\theta_1) \tag{45}$$

$$G_t(\theta_1, \theta_2) \succeq (1 + 2K \cdot (S_1 - S_2))^{-1} H_t(\theta_2), \tag{46}$$

*where $K$ is defined in Eq. (36).*

*Proof.* Since $\{x^\top \theta : x \in \mathcal{X}, \theta \in \Theta\} \subset [S_2, S_1]$ by Assumption 1, we have

$$\sup\{\Gamma(x^\top \theta) : x \in \mathcal{X}, \theta \in \Theta\} \le K.$$

By Lemma 34, we have that for all $x \in \mathcal{X}$,

$$\alpha(x^\top \theta_1, x^\top \theta_2) \ge (1 + K|x^\top(\theta_1 - \theta_2)|)^{-1} \dot\mu(x^\top \theta_1) \ge (1 + 2K \cdot (S_1 - S_2))^{-1} \dot\mu(x^\top \theta_1).$$

Then the following holds

$$\sum_{i=1}^{t} \alpha(X_i^\top \theta_1, X_i^\top \theta_2) X_i X_i^\top \succeq (1 + 2K \cdot (S_1 - S_2))^{-1} \sum_{i=1}^{t} \dot\mu(x^\top \theta_1) X_i X_i^\top$$

$$G_t(\theta_1, \theta_2) \succeq (1 + 2K \cdot (S_1 - S_2)) H_t(\theta_1),$$

where the last inequality follows by $(1 + 2K \cdot (S_1 - S_2))^{-1} \le 1$. The proof of Eq. (46) follows by substituting $\theta_1$ with $\theta_2$. $\square$

**Lemma 30** ($\mathcal{C}_t^\delta(\hat\theta_t)$ has small ellipsoidal diameters). *Under Assumptions 1 and 2, for all $\theta_1, \theta_2 \in \mathcal{C}_t^\delta(\hat\theta_t)$, it follows that*

$$\|\theta_1 - \theta_2\|_{H_t(\theta_1)} \vee \|\theta_1 - \theta_2\|_{H_t(\theta_2)} \le 2(1 + 2K \cdot (S_1 - S_2))\gamma_t(\delta),$$

*where $K$ is defined in Eq. (36).*

*Proof.* We first prove the statement for $\|\theta_1 - \theta_2\|_{H_t(\theta_1)}$. By Lemma 37, we have that

$$
\begin{aligned}
\|\theta_1 - \theta_2\|_{H_t(\theta_1)} &\leq \sqrt{(1 + 2K \cdot (S_1 - S_2))} \|\theta_1 - \theta_2\|_{G_t(\theta_1,\theta_2)} \\
&= \sqrt{(1 + 2K \cdot (S_1 - S_2))} \|g_t(\theta_1) - g_t(\theta_2)\|_{G_t^{-1}(\theta_1,\theta_2)} \qquad \text{(Lemma 36)} \\
&\leq \sqrt{(1 + 2K \cdot (S_1 - S_2))} \left( \|g_t(\theta_1) - g_t(\hat{\theta}_t)\|_{G_t^{-1}(\theta_1,\theta_2)} + \|g_t(\hat{\theta}_t) - g_t(\theta_2)\|_{G_t^{-1}(\theta_1,\theta_2)} \right)
\end{aligned}
$$

Note that $\theta_1, \theta_2 \in \mathcal{C}_t^\delta(\hat{\theta}_t)$ by hypothesis, then Lemma 37 and the definition of $\mathcal{C}_t^\delta(\hat{\theta}_t)$ gives that

$$
\begin{aligned}
\|g_t(\theta_1) - g_t(\hat{\theta}_t)\|_{G_t^{-1}(\theta_1,\theta_2)} &\leq \sqrt{(1 + 2K \cdot (S_1 - S_2))} \|g_t(\theta_1) - g_t(\hat{\theta}_t)\|_{H_t^{-1}(\theta_1)} \\
&\leq \sqrt{(1 + 2K \cdot (S_1 - S_2))} \gamma_t(\delta) \\
\|g_t(\hat{\theta}_t) - g_t(\theta_2)\|_{G_t^{-1}(\theta_1,\theta_2)} &\leq \sqrt{(1 + 2K \cdot (S_1 - S_2))} \|g_t(\hat{\theta}_t) - g_t(\theta_2)\|_{H_t^{-1}(\theta_2)} \\
&\leq \sqrt{(1 + 2K \cdot (S_1 - S_2))} \gamma_t(\delta).
\end{aligned}
$$

Chaining all the inequalities together finishes the proof. The proof for the statement for $\|\theta_1 - \theta_2\|_{H_t(\theta_2)}$ follows similarly by substituting $\theta_1$ with $\theta_2$. $\square$

### E.6    Proof of self-bounding property of self-concordance functions: Lemma 31

As mentioned beforehand, the following lemma is abstracted out from Claim 14 of Janz, Liu, Ayoub, and Szepesvári [Jan+24]:

**Lemma 31** (Self-bounding property of self-concordance functions). *Let $\mathcal{V} = [a, b]$, a closed, nonempty interval over the reals, $f$ a real valued function defined over an interval of the reals that is twice continuously differentiable over $\mathcal{V}$ such that for some $\Gamma : \mathcal{V} \to \mathbb{R}_+$, $|\ddot{f}(v)| \leq \Gamma(v)\dot{f}(v)$ holds for all $v \in \mathcal{V}$. Assume that $A = \sup_{v \in \mathcal{V}} \Gamma(v) < \infty$. Furthermore, assume that either $\dot{f}$ is identically zero over $\mathcal{V}$, or $\dot{f}$ is positive valued over $\mathcal{V}$. For $n$ a positive integer, let $\{a_t\}_{t=1}^n \subset \mathcal{V}$. Then,*

$$
\sum_{t=1}^n \dot{f}(a_t) \leq n\dot{f}(b) + A \sum_{t=1}^n f(b) - f(a_t).
$$

*Proof.* We have

$$
\begin{aligned}
\sum_{t=1}^n \dot{f}(a_t) &= \sum_{t=1}^n \dot{f}(b) + \sum_{t=1}^n (a_t - b) \int_0^1 \ddot{f}(b + v(a_t - b)) \, dv \\
&\leq n\dot{f}(b) + \sum_{t=1}^n \left| (a_t - b) \int_0^1 \ddot{f}(b + v(a_t - b)) \, dv \right| \\
&\leq n\dot{f}(b) + \sum_{t=1}^n (b - a_t) \int_0^1 \left| \ddot{f}(b + v(a_t - b)) \right| dv \qquad (a_t \leq b \text{ and triangle inequality}) \\
&\leq n\dot{f}(b) + \sum_{t=1}^n (b - a_t) \int_0^1 \Gamma(b + v(a_t - b))\dot{f}(b + v(a_t - b)) \, dv \qquad \text{(Lemma 32)} \\
&= n\dot{f}(b) + \sum_{t=1}^n (b - a_t) \int_0^1 A\dot{f}(b + v(a_t - b)) \, dv \\
&\leq n\dot{f}(b) + K \sum_{t=1}^n f(b) - f(a_t), \qquad \text{(fundamental theorem of calculus)}
\end{aligned}
$$

finishing the proof. $\square$

### E.7 Auxiliary Lemma

**Lemma 38** (Elliptical potential lemma). *Fix $\lambda, A > 0$. Let $\{a_t\}_{t=1}^{\infty}$ be a sequence in $AB_2^d$ and let $V_0 = \lambda I$. Define $V_{t+1} = V_t + a_{t+1}a_{t+1}^{\top}$ for each $t \in \mathbb{N}$. Then, for all $n \in \mathbb{N}^+$,*

$$\sum_{t=1}^{n} \|a_t\|_{V_{t-1}^{-1}}^2 \leq 2d \max\left\{1, \frac{A^2}{\lambda}\right\} \log\left(1 + \frac{nA^2}{d\lambda}\right).$$

*Proof.* See, e.g., Lemma 19.4 of Lattimore and Szepesvári [LS20]. □

## F Numerical Simulations

In this section, we report our results on numerical simulations. We run our algorithm on exponential bandits. The setting is as follows. The base distribution is an exponential distribution with parameter $\lambda > 0$ where the probability density function can be written as

$$f(x; \lambda) = \mathbb{I}(x \geq 0)\lambda \exp(-\lambda x).$$

For each arm $x \in \mathcal{X}$, the reward distribution is an exponential distribution with parameter $\lambda - x^{\top}\theta_{\star}$, where we ensure that $\sup_{x \in \mathcal{X}} x^{\top}\theta_{\star} < \lambda$, as illustrated by Example 1. The number of arms $|\mathcal{X}| = 20$ and $\mathcal{X} \subseteq \mathbb{R}^2$, the max variance of the reward distributions among all arms is $0.25$ and $\kappa = \sup_{x \in \mathcal{X}} 1/\dot{\mu}(x^{\top}\theta_{\star}) \approx 100$. We run our algorithms with theory suggested parameters for 60 runs where each run has horizon 5000. To be more specific on the parameters, we list them here:

1. The failure probability $\delta$ is $0.05$.
2. The regularizer is set to be 2.
3. The confidence width $\gamma_t(\delta, \lambda)$ is set according to Eq. (43).

Here are the results of the experiments. In Fig. 1 we plot the average regret along with standard

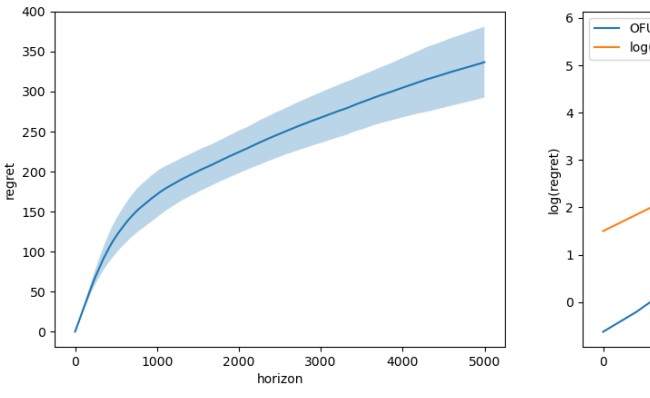
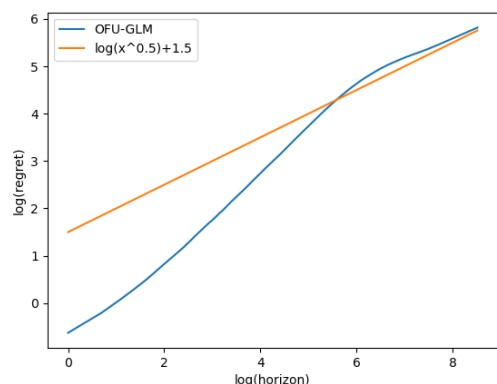

Figure 1: Average of 60 runs      Figure 2: The log-log plot of OFU-GLM

deviation. From the plot we can see that the regret attained seems to be sublinear. In Fig. 2 we display the log-log plot where the x-axis is $\log(\text{horizon})$ and the y-axis is $\log(\text{regret})$. The slope gradually approaches to 0.5, i.e., the growth rate of regret approaches to $\sqrt{T}$ which confirms our theoretical bound.

