# OpenReview forum: "Almost Free: Self-concordance in Natural Exponential Families and an Application to Bandits"
_NeurIPS.cc/2024/Conference — NeurIPS 2024 poster_

### Official Review · Reviewer_ySJz · 2024-07-08

**Soundness:** 3
**Presentation:** 1
**Contribution:** 3
**Rating:** 5
**Confidence:** 5

**Summary:**

This paper discusses Generalized Linear Bandits (GLB) under the self-concordance assumption. The study successfully relaxes the limitations of existing work with an OFU-type algorithm, providing mathematically solid theories for GLB within the self-concordance family. However, the presentation could be significantly improved, and the paper's style does not align well with the venue.

**Strengths:**

This paper has a sound theoretical foundation with the relaxation of some assumptions in the existing literature.

**Weaknesses:**

The first section should provide a comprehensive overview of the study. The core concept of "self-concordance" is introduced abruptly and is not well-explained. The literature review is incomplete. Specifically, the differences between this study and the existing literature, such as [Rus+21], are not thoroughly discussed. As a result, the contribution of this study is unclear. The paper dedicates the first seven pages to the theoretical foundation and introduces the model and algorithm thereafter. Due to space constraints, these are not fully illustrated, and the paper lacks experimental results. This organization is not well-suited to this venue.

**Questions:**

This paper talks about "being free of exponential dependency". I would like to see what the term means in detail. Also, why "being free of exponential dependency" is important in analyses and applications?

**Limitations:**

This paper discusses the single-parameter natural exponential family. But, I believe that this family is not comprehensive enough to include most applications in the real world.

---

> ### Author Rebuttal · Authors · 2024-08-06
>
> We would like to clarify a misunderstanding of our work that we do not “discuss GLB under the self-concordance assumption”. What is true that our work goes beyond this: we aim to remove this assumption for GLBs with subexponential base distribution.
>
> Now we would like to address concerns pointed out in the weakness section.
> + Due to the space limitation, we are unable to provide a thorough explanation for self-concordant functions but we would like to kindly remind the reviewer that references of its origin are provided in line 127 and we introduce its use in machine learning & optimization from line 28 to 46. In the future versions of our manuscript, we plan on using the extra page to flesh out the exposition around self-concordance as well strengthening the section on related works.
> + There is a huge body of related work, so we are incapable of summarizing them all thoroughly due to the lack of space (which we think is a common practice, in the exploding field we are in). All the related works that we know of are at least mentioned. On the issue of why [Rus+21] is not discussed in detailed. The main contribution in this paper is working out how to act in a non-stationary setting. Since we only consider the stationary setting, their contribution is irrelevant from the perspective of our paper (they avoid any difficulty that we face by assuming bounded rewards). Finally, the regret bound in [Rus+21] scales with $\kappa$ (which can be exponentially large in the dimension $d$, [8]) and is not second order (meaning the bounds does not scale with the optimal arm’s variance); the stronger earlier relevant papers are all discussed in our paper; we'd be happy to add more relevant works if needed and did the reviewer have any other suggestions.
> + We would like to note that our writing style naturally flows from theory to algorithms, as appreciated by reviewers Ca74 and n2uQ. The contribution of this work is to show that GLBs with subexponential base distribution possess self-concordance property. It is a result that could be interesting on its own and the application of it could go beyond GLBs as detailed in section 5. We select GLBs as a demonstration of how the self-concordance property of subexponential NEFs can be applied to address open problems in the bandit literature. Our results naturally extend the analysis techniques presented in [9]. This motivates the organization of our paper.
>
> ySjz asks the question about being free of exponential dependency. This originally comes from the very first work on GLBs [1] where the confidence width (and as a result, the regret bound) suffers from a dependency on $\kappa$ that is in worst case exponential in the norm of true underlying parameter $\theta_\star$, creating a significant gap between theoretical and empirical results. To explain why resolving exponential dependency is important, we would like to remind ySjz that algorithms with exponential dependency in relevant parameters are considered as inefficient and unpractical as the resource demands increase very rapidly with input size. Transferring this notion into decision making, or specifically bandit algorithms, algorithms with regret scaling exponentially with relevant parameters are considered to make inefficient use of samples. As is pointed out in [8], which studies an instance of GLB -  logistic bandit: *“$\kappa$ can be prohibitively large, which drastically worsens the regret guarantees as well as the practical performances of the algorithms.”* (paragraph **Limitations**) and *“Even for reasonable values of $S$, this has a disastrous impact on the regret bounds”* (section 2). Removing this exponential dependency enables us to align the theoretical bound of UCB-type algorithms on GLBs with its empirical performance. To the best of the authors’ knowledge, there is no previous work resolving this exponential dependency for the whole class of GLBs with subexponential base distribution.
>
> We would like to address the concern about the applicability of the work. We are confused by the reviewer’s claim that natural exponential families are “not comprehensive enough to include most applications in the real world”. We interpret this sentence as asking whether NEFs enjoy revenue generating applications in the real work, which they do. To give a concrete example, consider Poisson regression, which is widely used when dealing with count data, e.g. number of insurance claims or Uber's surge pricing model.
>
> As for the theoretical nature of the work, we would like to point out that many important Neurips publications are theoretical in nature. Here is a non-exhaustive list of Neurips publications dedicated fully to theory without experimental results
> [2, 3, 4] and all of them are cited over 300 times. Here are works accepted last year that dedicate themselves fully to theory without experimental results [5,6,7].
> We also conducted some numerical experiments. The results are available in the global rebuttal.
>
> [1]. Filippi et al. Parametric bandits: The generalized linear case. In NeurIPS 2010\
> [2]. Jin et al. Is q-learning provably efficient? In NeurIPS 2018\
> [3]. Kakade et al. On the Complexity of Linear Prediction: Risk Bounds, Margin Bounds, and Regularization. In NeurIPS 2008\
> [4]. Antos et al. Fitted q-iteration in continuous action-space MDPs. In NeurIPS 2007\
> [5]. Liu et al. Optimistic natural policy gradient: a simple efficient policy optimization framework for online rl. In NeurIPS 2023\
> [6]. Yuan et al. Optimal extragradient-based algorithms for stochastic variational inequalities with separable structure. In NeurIPS, 2023\
> [7]. Foster et al. Model-free reinforcement learning with the decision-estimation coefficient. In NeurIPS 2023\
> [8]. Faury et al. Improved Optimistic Algorithms for Logistic Bandits. In ICML 2020\
> [9]. Abeille et al. “Instance-Wise Minimax-Optimal Algorithms for Logistic Bandits.” In AISTATS 2021\

---

> > ### Author Response · Authors · 2024-08-11
> >
> > Dear Reviewer,
> >
> > The author-reviewer discussion period is shortly coming to an end. We were hoping to get a confirmation from you that you've considered our rebuttal, and to let us know if you have any further questions.

---

> > > ### Comment · Reviewer_n2uQ · 2024-08-12
> > >
> > > As a fellow reviewer, I wanted to express my disappointment in this review, which is unfair, if not adversarial, in its assessment of the work. Overall, I feel that the reviewer doesn't have sufficient expertise to review this paper and acknowledge the significance of the technical contributions, despite the confidence 5.
> > >
> > > The notion of self-concordance property is essential in the analysis of generalized linear bandits, in which prior literature either focused on bounded GLMs or GLMs that are explicitly assumed to satisfy the self-concordance property. Thus, the criticisms of the "abrupt" introduction and unclear contributions are invalid. Although it would have been nice to include experiments from the beginning, the theoretical contributions deserve some merit. Those are not just "relaxation of some assumptions" but a significant relaxation in that the stretch factor needs not to be constant, and the other technical contributions on how to use such self-concordance to derive the *first* $d \sqrt{T/\kappa}$ regret bound over generic generalized linear bandits are also quite significant. I'm quite confused about what the reviewer means by "not well-suited to this venue," which is definitely not true. Moreover, saying that the single-param NEF is not comprehensive enough is just plain wrong.
> > >
> > > The review misses the whole point and context of this paper's contributions, which doesn't add much to the fair assessment of their soundness and relevance.

---

> > ### Comment · Reviewer_ySJz · 2024-08-14
> >
> > Thanks for the detailed explanation. I understand the authors' concerns as well as n2uQ's disappointment. I agree that my understanding of fitness in this venue can be wrong.
> >
> > But, I still believe that this paper has a high barrier for most readers and the presentation should be improved for readers without background knowledge about GLB.
> >
> > I think that the authors' rebuttals are valid. I will more carefully review your manuscript based on the rebuttal and make changes in my evaluation.

---

### Official Review · Reviewer_n2uQ · 2024-07-13

**Soundness:** 4
**Presentation:** 4
**Contribution:** 4
**Rating:** 8
**Confidence:** 3

**Summary:**

The authors prove that any single-parameter natural exponential family (NEF) with subexponential (subgaussian) base distribution is self-concordant with a stretch factor that grows inverse quadratically (linearly).

**Strengths:**

- Clearly and well-written
- An important theoretical contribution in establishing that self-concordance for exponential tilting comes for free, which has numerous applications in statistics and bandits (as alluded to in the Conclusion). Surprisingly, the proof consists of some concentration results combined with clever integral/infinite sum manipulations.
- For the subgaussian, the authors established that the linear growth is tight by establishing a lower bound, which is technically quite interesting.
- First $d\sqrt{T/\kappa_\star}$ type regret that holds for a wide range of generalized linear bandits, well beyond logistic bandits

**Weaknesses:**

- The confidence set for the OFU-GLB is nonconvex, and thus, the current algorithm is computationally intractable. There has been some progress on achieving convex confidence sets that are also statistically (regret-wise) tight -- see, e.g., [1,2]. It would be nice to write the confidence set in a convex form, as it is repeatedly alluded to during the authors' proof.
- With the convex confidence set, it would be nice to see some numerical experiments, especially the exponential bandits.
- The norm parameters $S_0, S_1, S_2$ were introduced but never kept track of. The dependencies on these norm parameters have been a subject of study on their own [2,3] and are known to perform well [2]. It would be nice to have those dependencies appearing in the main text as well.
- Most of the discussions regarding the proof of regret analyses are relegated to the Appendix. From my perspective (heavily biased towards bandits community), the techniques used there are arguably a bit more interesting. I would really like to see the technical contributions made in the main text's Section 4, especially the self-concordance control lemmas used (Appendix E.3) and Lemma 31 and more.


**Comments**
- It would be nice to include some discussions regarding [3], which provides the first $d\sqrt{T/\kappa_\star}$-type regret for *bounded* generalized linear bandits but still goes beyond logistic bandits.
- For each theorem statement, if not already done so, I would like to see a direct (hyper)ref to the Appendix containing its proof
- Typo: "Lemma ??" in pg. 24

[1] https://proceedings.mlr.press/v130/abeille21a.html

[2] https://proceedings.mlr.press/v238/lee24d.html

[3] https://arxiv.org/abs/2404.06831

**Questions:**

- Can the regret analyses be extended to changing arm-set case, i.e., the arm-set $\mathcal{X}_t$ varies across $t \in [T]$?
-

**Limitations:**

Yes

---

> ### Author Rebuttal · Authors · 2024-08-06
>
> Thanks for your review and suggestions and we really appreciate them!
> For the weakness section:
> + We agree that the convex relaxation technique from Abeille et al [1], can be applied to our case. We adapted the proof from Abeille et al [1] and are able to apply the convex relaxation to our confidence sets as well. This was a good suggestion and thus will be added to the future version of our manuscript.
> + We conducted some numerical experiments on exponential bandit (as suggested by the reviewer) to verify our theoretical contributions. The results are available in the global rebuttal.
> + We will make the dependency on $S_0$, $S_1$ and $S_2$ explicit in the regret bound. Thank you for drawing our attention to this issue.
> + The techniques in section 4 follow similarly to Abeille et al [1]. Lemma 32 is borrowed from proposition 8 of Sun and Tran Dinh [2], where we tailored the technique into our specific setting. Similar use of self-concordance is also present in [1,3].
>
> For the comment section:
> + We will be happy to discuss [4] in the final version. Their arm elimination technique and adaptation of rare switching are relevant. It is also nice to see that their empirical performance can surpass the performance of ECOLog, which, according to our experience, is quite competitive.
> + We will be happy to add hyperlinks to the appendix about the proofs of each statement.
> + Thanks for pointing out the typo. It was meant to be the lower bound on $\dot\mu()$, i.e., lemma 17.
>
> For the question section: Yes, our regret guarantee still holds for changing arm-sets as long as  $S_2\le x^T\theta\le S_1$ holds for all $x\in \mathcal X_t$ , $\theta\in \Theta$ and $t\in [T]$. Our proof technique makes no use of the stationarity of the arm set and its inclusion was solely to facilitate a more streamlined exposition.
>
> [1]. Marc Abeille, Louis Faury, Clement Calauzenes. “Instance-Wise Minimax-Optimal Algorithms for Logistic Bandits.” In International Conference on Artificial Intelligence and Statistics (2021).
>
> [2]. Tianxiao Sun and Quoc Tran-Dinh. “Generalized self-concordant functions: a recipe for Newton-type methods.” In Mathematical Programming 178 (2017): 145 - 213.
>
> [3]. David Janz, Shuai Liu, Alex Ayoub and Csaba Szepesvari. “Exploration via linearly perturbed loss minimisation.” In International Conference on Artificial Intelligence and Statistics (2024).
>
> [4]. ​​Ayush Sawarni, Nirjhar Das, Siddharth Barman, Gaurav Sinha. "Generalized Linear Bandits with Limited Adaptivity." ICML 2024 Workshop: Aligning Reinforcement Learning Experimentalists and Theorists.

---

> > ### Comment · Reviewer_n2uQ · 2024-08-12
> >
> > Thank you for the responses, and apologies for getting back so late.
> >
> > After reading through the responses to my and other reviewer's reviews, I'm satisfied with the authors' responses and intend to keep my score.

---

### Official Review · Reviewer_Ca74 · 2024-07-13

**Soundness:** 3
**Presentation:** 3
**Contribution:** 3
**Rating:** 6
**Confidence:** 2

**Summary:**

The paper investigates the self-concordance properties of single-parameter natural exponential families (NEFs) with subexponential tails. It provides two main contributions: first, it demonstrates that NEFs with subexponential tails are self-concordant with polynomial-sized parameters, and second, it applies these findings to generalized linear bandits (GLBs). The authors derive novel second-order regret bounds for GLBs that are free of exponential dependence on the problem parameters. This result extends the applicability of optimistic algorithms for GLBs to reward distributions such as Poisson, exponential, and gamma.

**Strengths:**

- The results on the self-concordance property of NEFs are definitely interesting and general, going beyond its application to GLBs. They can be interesting to many other problem settings (as mentioned by the authors themselves).
- Not only do the authors provide an algorithm for GLBs, but it guarantees a second-order regret bound. This is particularly desirable to achieve better data adaptivity.
- The results are presented in a clear manner, with clear definitions and helpful examples.

**Weaknesses:**

- It appears that the algorithmic results in this work tightly rely on the knowledge of some parameters of the NEF. Parameter-free results are quite important and have become relevant, especially in the past few years.
- The paper could benefit from a slightly more detailed comparative analysis with existing methods, highlighting the advantages and potential trade-offs of the proposed approach in various scenarios, and how these prior results would fit in the setting considered in this work.
- The papers could benefit from experimental results, although it is very clear that this is a theoretical paper.

Minor details:
- Many references have "et al." instead of the full list of authors. This should be fixed.

**Questions:**

- Do the authors believe it possible to lift prior knowledge on parameters such as $S_0$, $L$, $c_1$, and $c_2$ in designing efficient algorithms for GLBs with NEF rewards?

**Limitations:**

The authors addressed potential limitations.

---

> ### Author Rebuttal · Authors · 2024-08-06
>
> Thanks for reviewing our manuscript and we are delighted at your appraisal of our work. A question was asked “Do the authors believe it possible to lift prior knowledge on parameters such as $S_0$, $L$, $c_1$ and $c_2$ in designing efficient algorithms for GLBs with NEF rewards?”
>
> To the best of the authors’ knowledge, existing parameter free algorithms on multi-armed bandits (linear bandits) still require the knowledge of subexponential (subgaussian) parameters or the upper bound of the reward [1], [2], [3]. Hence the knowledge of subexponential parameters (i.e., $c_1$, $c_2$) may be still needed. The knowledge of $S_0$ can be lifted as the information we need about $S_0$ can be provided by $S_1$ and $S_2$. Judging from the present work on parameter-free bandit algorithms [3], the knowledge of $S_1$ and $S_2$ cannot be lifted as [3] designs their confidence set using the upper bound of the true underlying model parameter $\theta_\star$ which plays a similar role to $S_1$ and $S_2$ in our work. The knowledge of $L$ can be lifted as it is worst-case polynomial in $\max(1/(c_1-S_1), 1/(c_2-S_2))$. This bound is oftentimes pessimistic hence we leave the dependency on $L$ in the bound and knowing a tighter bound on the variance definitely helps in the performance. Hence, removing dependencies on knowledge of subexponential/subgaussian parameter and knowledge of the bound of the parameter norm, $S$, are promising directions for future research and remain open in the bandit literature.
>
> We now address concerns in the weakness section.
> + For more detailed comparative analysis, we can compare our bandit algorithm with subgaussian base distribution and we will be happy to add it to the camera ready version of the manuscript (assuming the paper gets accepted). Specifically, we will point out the setting in this work is the most general in the sense that we consider GLBs with subexponential rewards while all the previous work we know of consider bounded rewards or subgaussian rewards. For subgaussian rewards, we will emphasize that [4,5,6,7] still depend on $\kappa$. Russac et al [7], also employs self-concordance in GLBs but they assume bounded reward and focuses on addressing non-stationarity of the environment – their bounds also scale with $\kappa$ in the leading term, which can be exponentially large for logistic bandits. Janz et al [8], consider a similar setting to ours. They assume the moment generating function of the base distribution $Q$ is defined over the entire real line. This implies that $Q$ does not have a tail as heavy as an exponential distribution hence less general than our setting.
> + For experiments, we provided some results in the global rebuttal.
> + For the minor details, thank you for drawing our attention to the formatting of our references! This will be addressed in future versions of our submission.
>
>
> [1]. Shinji Ito (2021). “Parameter-free multi-armed bandit algorithms with hybrid data-dependent regret bounds”. In COLT 2021\
> [2]. Yifang Chen, Chung-Wei Lee, Haipeng Luo, Chen-Yu Wei. “A new algorithm for non-stationary contextual bandits: efficient, optimal and parameter-free.” In COLT 2019\
> [3]. Kei Takemura, Shinji Ito, Daisuke Hatano, Hanna Sumita, Takuro Fukunaga, Naonori Kakimura, Ken-ichi Kawarabayashi. “A parameter-free algorithm for misspecified linear contextual bandits.” In AISTATS 2021\
> [4]. Sarah Filippi, Olivier Cappe, Aurélien Garivier, Csaba Szepesvári. Parametric bandits: The generalized linear case. In NeurIPS 2010.\
> [5]. Kwang-Sung Jun, Lalit Jain, Blake Mason, Houssam Nassif. Improved Confidence Bounds for the Linear Logistic Model and Applications to Bandits. In ICML 2021.\
> [6]. Aadirupa Saha, Aldo Pacchiano, Jonathan Lee. Dueling RL: Reinforcement Learning with Trajectory Preferences. In AISTATS 2023\
> [7]. Yoan Russac, Louis Faury, Olivier Cappé, Aurélien Garivier. Self-Concordant Analysis of Generalized Linear Bandits with Forgetting. In AISTATS 2021.\
> [8]. David Janz, Shuai Liu, Alex Ayoub and Csaba Szepesvari. “Exploration via linearly perturbed loss minimisation.” In AISTATS 2024.\

---

> > ### Comment · Reviewer_Ca74 · 2024-08-14
> >
> > I thank the authors for the response. I have no further questions and am keeping my positive score.

---

### Author Rebuttal · Authors · 2024-08-06

We would like to thank you all for the time and effort spent in reviewing our manuscript! We are glad that the reviewers appreciate our theoretical contributions on NEFs (reviewer Ca74 & reviewer n2uQ), our contributions to the bandit literature on second-order regret guarantee for GLBs with subexponential rewards (reviewer Ca74 & reviewer n2uQ) and our writing style (reviewer Ca74 & reviewer n2uQ). A general concern was raised about the lack of experiments and we understand the concern and conducted some experiments -- as a sanity check of our theoretical results. The setting of the experiment is as follows. We will be happy to use the extra space in the paper for including a little more thorough experimental verification of our theoretical results.
+ We conduct experiment on exponential bandits, i.e., the base distribution is an exponential distribution and as a result the reward distributions are also exponential.
+ The dimension $d$ of the true underlying parameter $\theta_\star$ is $d=2$
+ The number of arms $|\mathcal X|=20$.
+ The maximum variance is $0.25$, i.e., for all $t\ge 1$, $\max_{x\in \mathcal X}\mathrm{Var}[Y_t|X_t=x]=0.25$ .
+ The experiment consists of 60 runs with a horizon of 5000.
+ The failure probability $\delta$ is set to be 0.05.
+ The regularizer is set to be $\lambda=2$ and $\gamma_t(\delta)$ to be the theory suggested value below line 922:
$$\gamma_t(\delta) = \sqrt{\lambda}\left(\frac{1}{2M}+S_0\right)+\frac{2Md}{\sqrt{\lambda}}\left(1+\frac{1}{2}\log\left(1+\frac{tL}{\lambda d}\right)\right)+\frac{2M}{\sqrt\lambda}\log(1/\delta)$$
The results are available in the pdf attached. Here are some explanations:
+ The first figure displays the mean regret along with standard deviation. The regret attained appears to be "sublinear".
+ The second figure is a log-log plot to display the growth rate of the regret. The slope gradually approaches 0.5, i.e., the growth rate of regret approaches to $\sqrt T$, which confirms our theoretical bound.

---

### Decision · Program_Chairs · 2024-09-25

**Decision:**

Accept (poster)

**Comment:**

This paper provides a general study of self-concordance in NEF, with applications to regret bounds for generalized linear bandits. Compared to previous works, which primarily focus on the logistic bandit, the paper generalizes the family of distributions for which self-concordance holds.

Overall the paper was well written. I agree with the reviewers that additional discussion and experiments comparing with the existing literature would be enlightening in the camera ready. In addition, highlighting some of the proof techniques in the main text. The authors may also want to discuss a concurrent work (https://arxiv.org/pdf/2407.13977) and highlight some of the differences.